# p53 promotes revival stem cells in the regenerating intestine after severe radiation injury

Clara Morral[1,14], Arshad Ayyaz[2,13,14], Hsuan-Cheng Kuo[3,14], Mardi Fink[2,4], Ioannis I. Verginadis[1], Andrea R. Daniel[5], Danielle N. Burner[3], Lucy M. Driver [5,6], Sloane Satow[5], Stephanie Hasapis[5], Reem Ghinnagow[1], Lixia Luo[5], Yan Ma[5], Laura D. Attardi [7], Constantinos Koumenis[1], Andy J. Minn [1,8,9,10], Jeffrey L. Wrana [2,4] ✉, Chang-Lung Lee [5,6] ✉ & David G. Kirsch [3,5,11,12] ✉

Ionizing radiation induces cell death in the gastrointestinal (GI) epithelium by activating p53. However, p53 also prevents animal lethality caused by radiation-induced acute GI syndrome. Through single-cell RNA-sequencing of the irradiated mouse small intestine, we find that p53 target genes are specifically enriched in regenerating epithelial cells that undergo fetal-like reversion, including revival stem cells (revSCs) that promote animal survival after severe damage of the GI tract. Accordingly, in mice with p53 deleted specifically in the GI epithelium, ionizing radiation fails to induce fetal-like revSCs. Using intestinal organoids, we show that transient p53 expression is required for the induction of revival stem cells and is controlled by an Mdm2-mediated negative feedback loop. Together, our findings reveal that p53 suppresses severe radiation-induced GI injury by promoting fetal-like reprogramming of irradiated intestinal epithelial cells.

Radiation tolerance of the gastrointestinal (GI) tract limits the effectiveness of radiation therapy for thoracic, pelvic, or abdominal malignancies, such as pancreatic cancer. The radiation tolerance of the GI tract can also be exceeded in radiation accidents at nuclear power plants or after the detonation of a nuclear weapon or radioactive bomb, which can cause a lethal radiation-induced GI syndrome[1]. There are currently no treatments to mitigate radiation-induced GI syndrome that have been approved by the US Food and Drug Administration[1].

Thus, there is a critical need to understand the mechanisms of radiation-induced GI injury and regeneration.

During homeostasis, epithelial cell turnover in the intestine is maintained by Lgr5[+] crypt base columnar (CBC) cells located at the bottom of intestinal crypts, which self-renew and give rise to differentiated progeny that constitute the intestinal epithelium[2]. The intestinal epithelium can fully recover from acute injury after a single radiation dose that ablates the Lgr5[+] CBCs. However, continuous Lgr5[+]

[1]Department of Radiation Oncology, Perelman School of Medicine, University of Pennsylvania, Philadelphia, PA, USA. [2]Centre for Systems Biology, Lunenfeld-Tanenbaum Research Institute, Mount Sinai Hospital, Toronto, ON, Canada. [3]Department of Pharmacology and Cancer Biology, Duke University, Durham, NC, USA. [4]Department of Molecular Genetics, University of Toronto, Toronto, ON, Canada. [5]Department of Radiation Oncology, Duke University, Durham, NC, USA. [6]Department of Pathology, Duke University, Durham, NC, USA. [7]Departments of Radiation Oncology and Genetics, Stanford University, Palo Alto, CA, USA. [8]Abramson Family Cancer Research Institute, Perelman School of Medicine, University of Pennsylvania, Philadelphia, PA, USA. [9]Parker Institute for Cancer Immunotherapy, Perelman School of Medicine, University of Pennsylvania, Philadelphia, PA, USA. [10]Mark Foundation Center for Immunotherapy, Immune Signaling, and Radiation, University of Pennsylvania, Philadelphia, PA, USA. [11]Radiation Medicine Program, Princess Margaret Cancer Centre, University Health Network, Toronto, ON, Canada. [12]Departments of Radiation Oncology and Medical Biophysics, University of Toronto, Toronto, ON, Canada. [13]Present address: Department of Biological Sciences, University of Calgary, Calgary, AB, Canada. [14]These authors contributed equally: Clara Morral, Arshad Ayyaz, Hsuan-Cheng Kuo. ✉e-mail: wrana@lunenfeld.ca; chang-lung.lee@duke.edu; david.kirsch@uhn.ca

CBC depletion leads to crypt loss and subsequent regeneration failure[3,4], indicating that Lgr5+ CBCs must be replenished after injury to repair the intestinal epithelium. Accordingly, recent studies support a paradigm in which the progeny of Lgr5+ CBCs, after chemical and radiation injury, undergo de-differentiation to reconstitute fresh Lgr5+ CBCs and regenerate the epithelium[5–8].

To investigate key signaling pathways that control the reversion of differentiated intestinal epithelial cells into stem cells following severe radiation injury, we focus on the tumor suppressor p53. We previously showed that p53 functions in the GI epithelium to prevent radiation-induced GI syndrome independent of apoptosis[9]; however, the underlying mechanism of this phenomenon remains incompletely understood. Here, we performed single-cell RNA-seq (scRNA-seq) and lineage tracing experiments to demonstrate that transient activation of p53 is required to properly reprogram damaged epithelial cells in response to severe radiation injury to promote tissue regeneration.

## Results

### Changes in epithelial cell populations following severe radiation injury in p53 wild-type mice

To examine how the loss of p53 affects cell states in the irradiated intestine, we performed scRNA-seq on crypt cells of the small intestines from mice that either retained functional p53 ($Villin^{Cre}$; $p53^{FL/+}$) or had p53 completely deleted ($Villin^{Cre}$; $p53^{FL/-}$) in the intestinal epithelium 2 days after 12 Gy sub-total-body irradiation (SBI) or sham irradiation. We first generated a reference map of the transcriptional response of the regenerating intestine with intact p53. We integrated the scRNA-seq data from mice harboring wild-type p53 ($Villin^{Cre}$; $p53^{FL/+}$) unirradiated or 2 days after irradiation with our previous scRNA-seq crypt data generated from wild-type mice unirradiated or 3 days after 12 Gy X-rays[5] (Fig. 1a). Different cell populations including hematopoietic cells, endothelial cells, fibroblasts, and epithelial cells were distinguished based on the expression of cell-specific marker genes. The epithelial cells that expressed Epcam were extracted, re-clustered, and used for downstream analyses (Supplementary Fig. 1a).

Unsupervised graph clustering identified 18 distinct groups of cells with different transcriptional profiles (clusters 0–17) (Supplementary Fig. 1b, and Supplementary Data 1). Each cluster was associated with a distinct intestinal cell type based on the expression of lineage-specific gene markers (Fig. 1b and Supplementary Fig. 1c). Enteroendocrine, Tuft cells, and Paneth cells were identified as distinct clusters by the expression of ChgA, Dclk1, and Defa22, respectively (Supplementary Fig. 1c), while absorptive enterocytes clusters were identified by the expression of Krt20 and Alpi and were divided into five distinct subclusters (Z1, Z2, Z3, Z4 and Z5) based on the expression of a previously described villus zonation gene signature[10] (Supplementary Fig. 1c, d and Supplementary Data 2). We also identified two distinct clusters of crypt base columnar cells (CBC1 and 2) that expressed canonical stem cell genes Lgr5, Smoc2, and Ascl2, and an adjacent cluster of transit amplifying (TA) cells marked by high expression of cell cycle genes (Supplementary Fig. 1c). We also interrogated the expression of the classical "+4" -also known as Label retaining cells (LRC)- gene signature (Bmi1, Tert, Hopx, Lrig1) because these genes have been previously reported as an alternative stem cell population located at position +4 from the intestinal crypt[11–14]. However, we did not detect specific enrichment of the LRC gene signature in any of the identified epithelial cells in our data set (Supplementary Fig. 1e).

By comparing clusters of epithelial cells from mice with and without irradiation, we noted two clusters of undifferentiated cells, clusters 10 and 14, were largely absent in unirradiated intestines. However, these two cell populations significantly emerged and expanded on days 2 and 3 after irradiation (Fig. 1c). Cells in cluster 14 expressed marker genes including Clu, Areg, Anxa2 associated with revival stem cells (revSCs) (Supplementary Fig. 1c, f), a unique

population of cells that is de-differentiated from the recent progeny of Lgr5+ CBCs and can regenerate the intestinal epithelium in response to radiation injury[5,15]. In addition, revSCs also strongly expressed the fetal-like gene Ly6a (Fig. 1d and Supplementary Fig. 1c, f), which is associated with the regeneration of injured intestinal epithelium[16,17]. Cells in cluster 10 also expressed Ly6a, while the expression of Clu in these cells was much lower compared to revSCs. Notably, these fetal-like undifferentiated cells also expressed certain CBC markers, such as Olfm4[18] and Ascl2[19] (Supplementary Fig. 1c). Therefore, we named the cells in cluster 10 fetal-like CBC cells (FCCs). Examination of cell proliferation markers Mki67 and Pcna indicated that revSCs are largely quiescent, while FCCs are highly proliferating (Fig. 1d and Supplementary Fig. 1g). Together, our findings from scRNA-seq revealed dynamic changes in the populations of intestinal epithelial cells during phases of radiation injury and regeneration—while 12 Gy SBI caused a substantial decrease in CBCs and TA cell populations, it induced the emergence of two fetal-like epithelial cell clusters that we identified as FCCs and revSCs, respectively (Supplementary Fig. 1h).

We validated the results from scRNA-seq using single-molecule fluorescence in situ hybridization (smFISH) in the intestinal tissues of p53 wild-type mice exposed to 12 Gy SBI. We observed that Ly6a was expressed by more than 70% of Clu+ cells, whereas only about 25% of Ly6a+ cells expressed Clu (Fig. 1e, f). In addition, smFISH results also showed that the expression of a cell cycle marker gene Mki67 was markedly higher in Ly6a+ cells compared to Clu+ cells (~50% versus 20%, respectively; Supplementary Fig. 1i, j). These results are consistent with our previous report that the Clu+ revSC cluster lacked the expression of proliferative makers[5]. Remarkably, smFISH assays detecting Clu and Ly6a mRNA showed a time-dependent change in the expression of these genes in the small intestines of p53 wild-type mice following 12 Gy. While Clu+ or Ly6a+ intestinal epithelial cells started to emerge as early as 48 hours after irradiation, the percentage of these cell populations was significantly increased at 60 and 72 hours after irradiation compared to unirradiated controls (0 hour) (Fig. 1g and Supplementary Fig. 1k). Together, these results show that the reprogramming of the intestinal epithelium after irradiation contains two distinct populations of fetal-like undifferentiated cells that are distinguished by their proliferative status: Ly6a+; Clu^{low/-} proliferating FCCs and Ly6a+; Clu+ quiescent revSCs.

### p53 promotes revSC-mediated regeneration of irradiated intestinal epithelial cells

It has been demonstrated that the emergence of Clu+ revSCs in intestinal epithelial cells following acute radiation injury is induced by several key signaling nodes, including TGFβ1[20] and the Hippo pathway effector protein Yes-associated protein (Yap)[5,15]. Indeed, we examined the transcriptional targets of Yap and found that these genes are also enriched in the revSC population after 12 Gy SBI (Supplementary Fig. 2a). Notably, in addition to Yap target genes, we also observed a significant enrichment of the p53 signature in FCCs and revSCs compared with CBCs and TAs (Fig. 2a, b and Supplementary Data 2). This finding was further supported by the fact that the expression of individual transcriptional targets of p53, including Phlda3, Cdkn1a, Bax, and Atg9b[21] was markedly increased in FCCs and revSCs in response to 12 Gy SBI (Fig. 2c). These results led us to speculate that p53 plays a crucial role in regulating fetal-like reprogramming during intestinal regeneration in response to severe radiation injury. To further investigate the potential relationship between Yap and p53 signaling, we revisited a scRNA-seq data set published by Cheung et al., 2020[22]. In this study, the authors deleted the Lats1/2 kinases in the mouse intestinal stem cells to induce Yap hyperactivation. Using this data set, we examined the expression of revSCs gene markers in addition to well-known transcriptional targets of p53. We found that hyperactivation of Yap via Lats1/2 knockout upregulated both sets of genes in intestinal stem cells, supporting the notion that Yap signaling

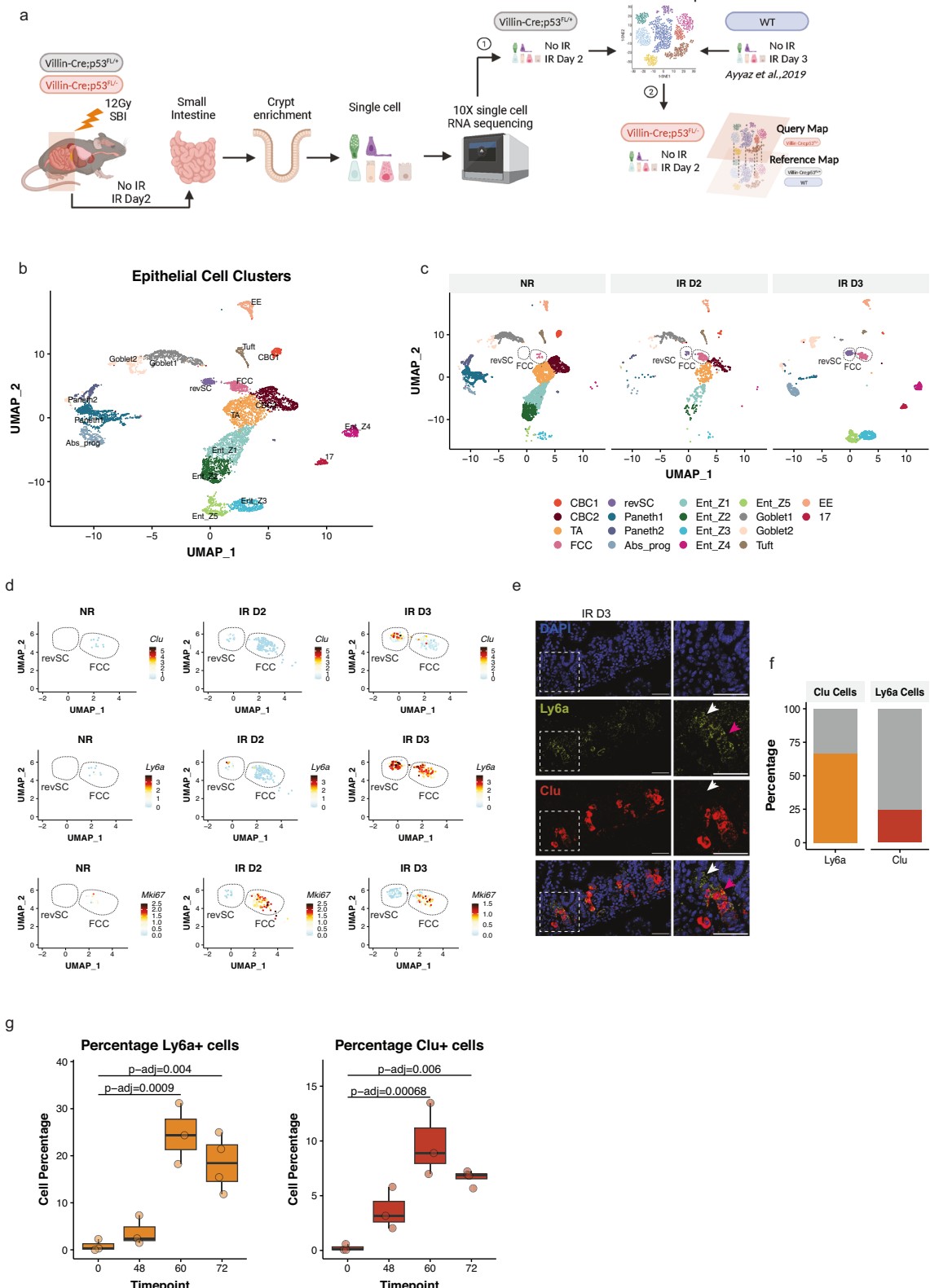

cooperates with p53 pathway activation to induce revSCs (Supplementary Fig. 2b, c).

To assess how p53 loss affects the intestinal epithelial response to severe radiation damage, we then used our reference map to query the cell clusters of *Villin^Cre^; p53^FL/–* mice at 2 days after 0 or 12 Gy (Fig. 3a). Strikingly, in contrast to mice that retained p53 (p53 WT), the emergence of revSCs and FCCs 48 hours after 12 Gy was impaired in mice

lacking p53 in the GI epithelium (p53 KO) (Fig. 3b). These results were validated by smFISH showing that the time-dependent induction of Clu^+; Ly6a^+ fetal-like revSCs following 12 Gy SBI was significantly inhibited in *Villin^Cre^; p53^FL/–* mice compared with *Villin^Cre^; p53^FL/+* mice (Fig. 3c, d).

To determine the impact of acute p53-inhibition after irradiation on the regeneration of intestinal crypts via Clu^+ revSCs, we established

**Fig. 1 | Revival stem cells and fetal-like CBC cells are induced after irradiation.**
**A** Workflow schematic for profiling the single-cell transcriptome of regenerating intestinal epithelium and generating a reference and query map. The reference map was generated using intestinal epithelial cells from p53 WT mice (p53[FL/+] and p53 WT) at indicated time points after 12 Gy SBI (step 1). The reference map was used to query the transcriptional profile of intestinal epithelial cells from p53[FL/−] mice (step 2). (IR: irradiated). (Created with BioRender.com). **B** UMAP representation of the reference map, including all time points with intestinal epithelial cell clusters labeled based on their transcriptional profile. **C** UMAP plots showing changes in cellular clusters between non-irradiated (NR) and two or three days after 12 Gy SBI (IR D2/D3), highlighting the emergence and expansion of revival stem cells (revSC) and fetal-like CBC cells (FCC) clusters after radiation exposure. **D** UMAP plots showing the mRNA expression pattern of *Ly6a, Clu,* and *Mki67* in revSC and FCC

clusters at different time points after SBI. **E** Representative smFISH images of Ly6a and Clu expression in mouse intestinal tissue sections at day 3 post-SBI, with white and pink arrowheads indicating Ly6a+Clu- and Ly6a+Clu+ cells, respectively (scale bar = 50 μm). Data from one experiment (*n* = 4 mice, *n* = 15–20 intestinal tissue areas/mouse). **F** Percentage of double Ly6a and Clu positive cells within the Clu (orange bar) and Ly6a (red bar) cell populations. Data represents quantification of 15 tissue sections across 4 mice (*n* = 4). **G** Percentage of *Ly6a* and *Clu* positive cells in intestinal tissue sections from NR mice (0 h) (*n* = 3) or 48 h (*n* = 3), 60 h (*n* = 3) and 72 h (*n* = 4) after 12 Gy of SBI. The boxplot represents the interquartile range (IQR) of the data, with the median indicated. The whiskers represent the highest and lowest values within 1.5 times the IQR. Each dot represents the average of 15 tissue areas quantified in each mouse. Statistical significance was calculated using a one-way ANOVA test followed by a Post hoc Tukey's HSD test.

a radiation injury model in intestinal organoids from *Clu[Cre-ERT2/+]; Rosa26[LSL-tdTomato/+]; Lgr5[DTR-GFP]* mice to enable in vitro lineage tracing experiments (Supplementary Fig. 3a). These organoids were treated with 4-hydroxytamoxifen (4-OHT) to label the Clu[+] revSCs and their offspring 2 hours before irradiation with 0, 3, or 5 Gy. Sixteen hours after irradiation, organoids were passaged and treated with either vehicle alone (DMSO) or pifithrin-alpha (PF-a), a small molecule inhibitor of p53[23]. Five days after passage, we examined the regeneration of Lgr5[+] cells labeled by GFP (green color) from the offspring of Clu[+] revSCs labeled by tdTomato (yellow color) (Fig. 3e, Supplementary Fig. 3b). In DMSO treated group, Lgr5[+] crypts derived from offspring of Clu[+] revSCs were 38% in unirradiated organoids but rose to >60% after 3 or 5 Gy. In contrast, PF-a treated organoids showed significantly decreased percentages of Lgr5[+]; tdTomato[+] crypts after 3 and 5 Gy (Fig. 3f, g). Of note, in Pf-a treated irradiated organoids tdTomato[+] cells (yellow) were mainly localized in the lumen. This phenomenon likely represents cells in which Clu was initially induced, but they were not properly reprogrammed to regenerate Lgr5[+] ISCs and subsequently repopulate the crypts in the absence of functional p53 (Fig. 3f, Supplementary Fig. 3a, b). These results suggest that p53 is required for key cell state transitions underlying intestinal regeneration in response to radiation injury, at least in part, by inducing and properly reprogramming Clu[+] revSCs.

We next examined whether Clu[+] revSCs are implicated in intestinal regeneration at higher radiation doses that cause GI syndrome by performing lineage tracing in previously described *Clu[Cre-ERT2/+]; Rosa26[LSL-tdTomato/+]* mice[5]. After 10 Gy or 14 Gy SBI, Clu[+] revSCs and their progeny were labeled with tdTomato by administering two doses of tamoxifen (TAM) at 1 and 2 days after irradiation (Supplementary Fig. 3c). In this mouse strain, 10 Gy and 14 Gy SBI are approximately LD$_{20/10}$ (the radiation dose that causes 20% of mice to develop GI syndrome within 10 days) and LD$_{50/10}$ doses for the development of radiation-induced GI syndrome, respectively (Supplementary Fig. 3d). Of note, we observed a significant increase in the percentage of Clu-labeled regenerated intestinal crypts after 10 Gy and 14 Gy compared to 0 Gy. In addition, crypts reconstituted through Clu[+] cells displayed a dose-dependent increase as radiation damage increased (50% after 14 Gy vs 25% after 10 Gy) (Supplementary Fig. 3e, f). Collectively, our results demonstrate that loss of functional p53 suppresses Clu[+] revSC-mediated regeneration of irradiated crypts in the small intestines in vivo and intestinal organoids in vitro. These findings reveal that p53-dependent induction of regenerative cell states promotes reconstitution of the small intestinal crypts after high-dose irradiation that causes GI syndrome.

### The p53-Mdm2 feedback loop is critical to control revSC-mediated intestinal regeneration
To examine the dynamics of p53 protein expression during epithelial recovery from radiation exposure, we co-stained p53 protein and Clu[+] cells at 0, 1, 2, or 3 days after 12 Gy using a GFP reporter driven by the Clu promoter (Clu[GFP]) that faithfully recapitulates endogenous Clu

expression in the small intestines[5]. We observed a significant induction of nuclear p53 protein at 2 days after 12 Gy, the timepoint when Clu[+] revSCs first start emerging in the irradiated intestine (Fig. 4a, b). However, the expression of p53 protein in intestinal epithelial cells was transient as it was substantially diminished by 3 days after irradiation (Fig. 4a, b and Supplementary Fig. 4a, b). The decrease in nuclear accumulation of p53 protein was associated with the induction of Mdm2, an E3 ubiquitin-protein ligase that degrades p53[24,25] (Fig. 4c and Supplementary Fig. 4c). Indeed, at 3 days post-irradiation, *Mdm2* mRNA was substantially induced in both revSCs and FCCs, but not in CBCs and TAs (Fig. 4c). The data from smFISH also confirmed a significant induction of Mdm2 mRNA in the intestinal epithelium 3 days after 12 Gy SBI (Fig. 4d, e). To investigate how the p53-Mdm2 negative feedback loop controls revSCs, we utilized the *Clu[Cre-ERT2/+]; Rosa26[LSL-tdTomato/+]; Lgr5[DTR-GFP]* organoid radiation damage model, and performed lineage tracing after irradiation as described above, but in this case treated with Nutlin-3, a small molecule Mdm2 inhibitor[26] (Fig. 4f). Continuous treatment of the organoids with Nutlin-3 after irradiation markedly increased p53 protein levels and decreased Clu-mediated crypt regeneration following 3 and 5 Gy (Fig. 4g and Supplementary Fig. 4d). In addition, Nutlin-3-treated irradiated organoids were smaller in size, had fewer buds, and did not survive past secondary passaging, indicating that the stem cell compartment was lost in the presence of continuous p53 activity (Fig. 4h). Together, our results reveal a critical role for transient p53 controlled by the p53-Mdm2 negative feedback loop in driving the Clu[+] revSC regenerative pathway after severe radiation injury (Supplementary Fig. 4e).

### The response of p53 to acute DNA damage is required to protect mice against radiation-induced GI syndrome
To define potential mechanisms by which p53 promotes proper regeneration of irradiated intestinal epithelium, we performed gene set enrichment analysis to identify signaling pathways significantly enriched after irradiation in crypts from mice with and without functional p53 (Supplementary Fig. 5a). Notably, we observed several pathways that were significantly upregulated in irradiated mice with intact p53, but they were downregulated after irradiation in mice that lost p53 in the GI epithelium. These pathways are mainly involved in the control of cell proliferation (Myc targets, oxidative phosphorylation, and E2F targets) or the DNA damage response (G2/M checkpoint and DNA repair) (Fig. 5a). The DNA repair pathway was also markedly enriched in both revSCs and FCCs (Fig. 5b). In addition, we observed significant upregulation of interferon alpha response after irradiation only in p53 deficient mice (Fig. 5a). These findings suggest that p53 may function as a transcription factor to have a pleiotropic impact on the regeneration of irradiated intestinal epithelium.

The transcriptional activity of p53 is controlled by two N-terminal transactivation domains (TAD): TAD1 and TAD2[21]. To dissect the role of p53-mediated transactivation in the radiation-induced GI syndrome, we conducted experiments using mice that harbor mutations in *p53* that impair either TAD1 alone or both TAD1 and TAD2. Previous studies

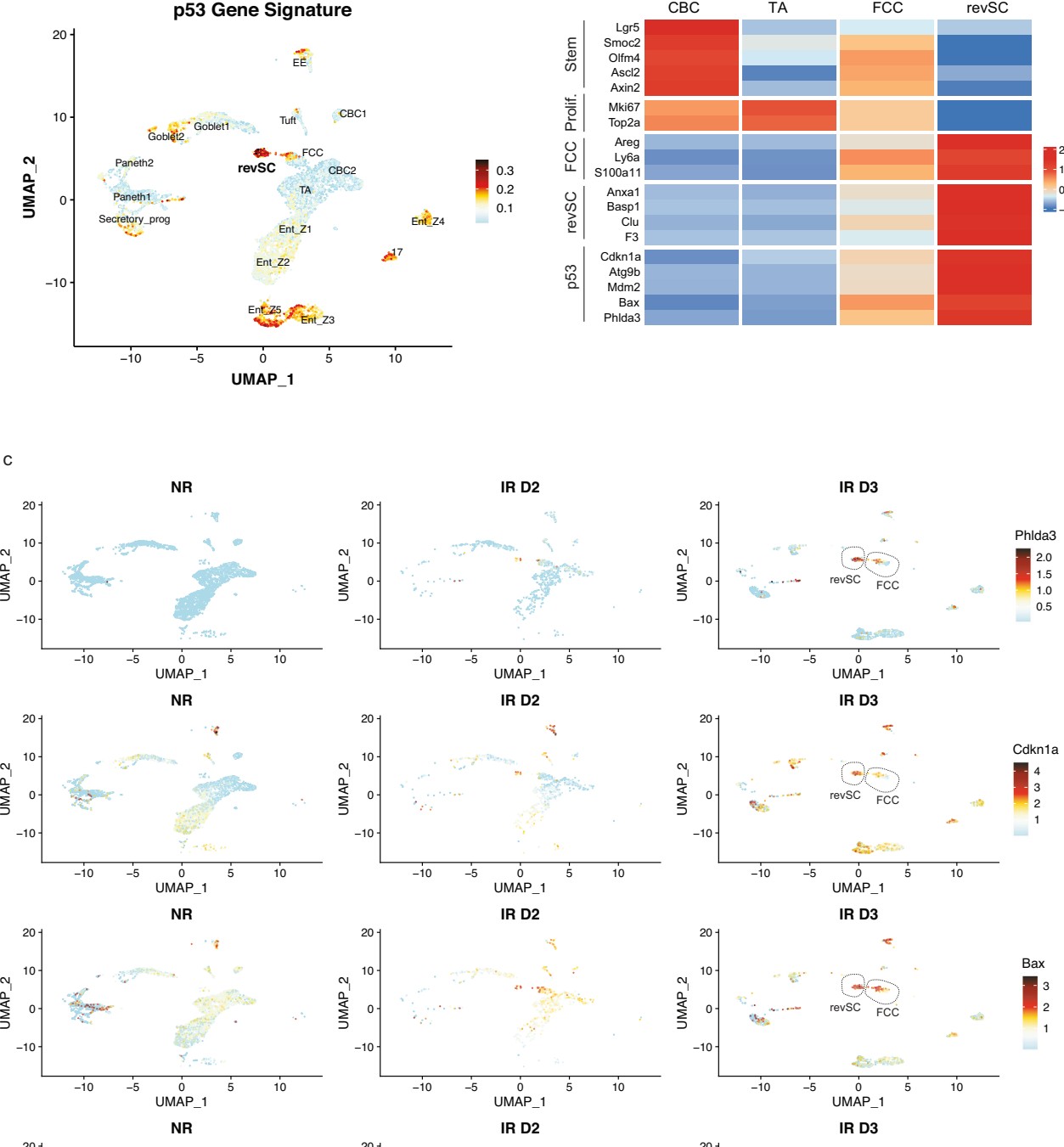

**Fig. 2 | p53 transcriptional targets are enriched in revSCs and FCC cells during regeneration. A** UMAP displaying the p53 transcriptional gene signature expression across epithelial intestinal cell clusters in p53 WT mice, including all time points. **B** Heatmap showing the average expression of selected gene markers across crypt base columnar cells (CBC), transit amplifying (TA), fetal-like CBC cells (FCCs), and revival stem cells (revSCs) cell populations. **C** UMAPS showing the expression of individual p53 transcriptional gene targets across all epithelial intestinal cell clusters in non-irradiated (NR) and 2 days (IR D2) and 3 days (IR D3) after 12 Gy SBI. RevSC and FCC clusters are highlighted at IR D3.

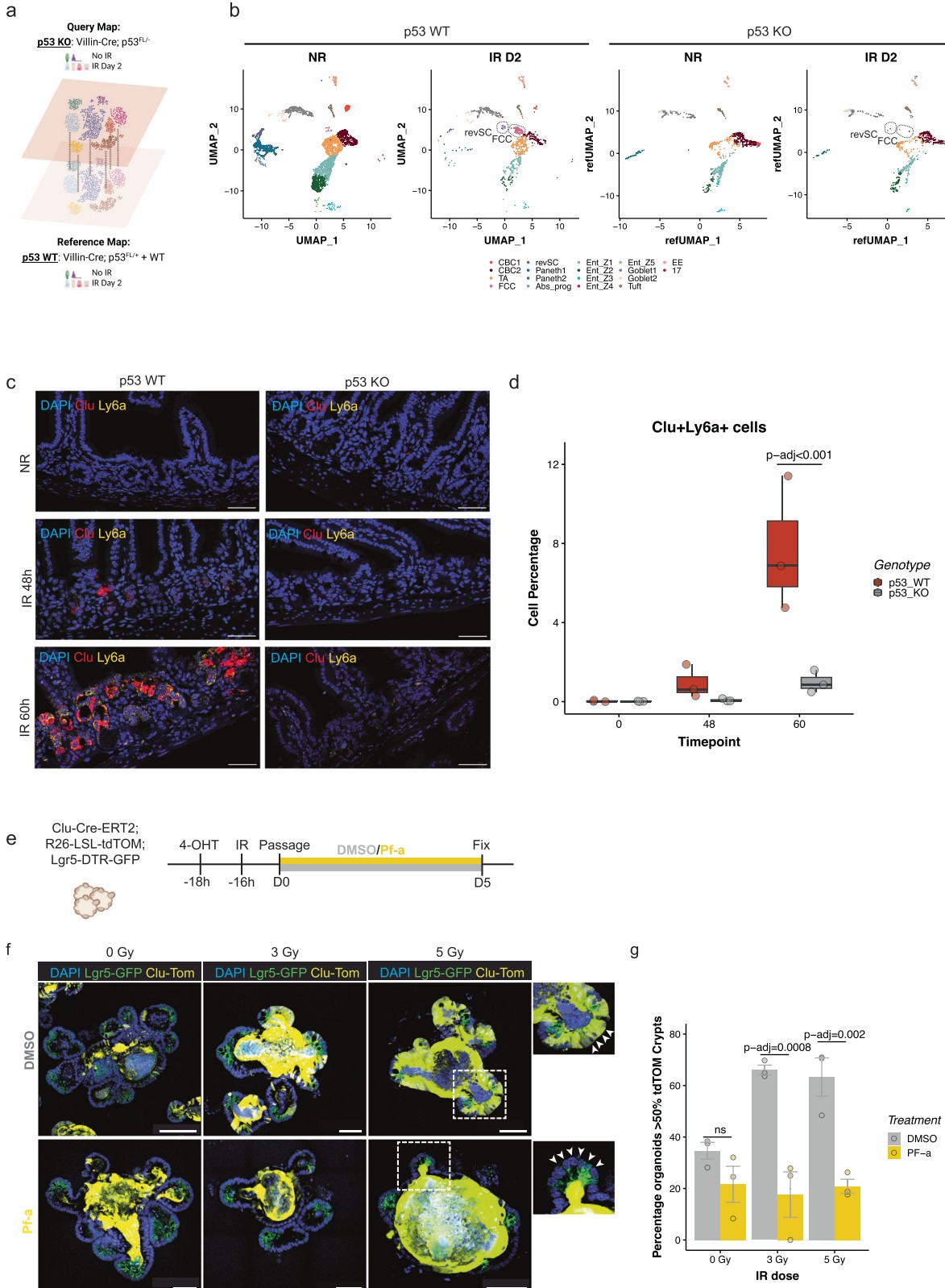

using these mice demonstrate that the transcription of genes that affect the canonical response of p53 to acute DNA damage is mainly dependent on TAD1, whereas the transcription of genes that mediate the non-canonical response of p53 is dependent on TAD1 and/or TAD2 in vivo[21]. Thus, we generated compound transgenic mice using *Villin^Cre* to delete one floxed p53 allele (FL) and remove a transcription/translation STOP cassette to activate expression of p53[25,26] with mutations in

TAD1 alone (*Villin^Cre*; *p53^LSL-25,26/FL*), which impairs the response of p53 to acute DNA damage. The same strategy was used to activate the expression of p53[25,26,53,54] with mutations in both TAD1 and TAD2 (*Villin^Cre*; *p53^LSL-25,26,53,54/FL*), which impairs all p53-dependent transcriptional responses (Fig. 5c). Immunohistochemistry (IHC) showed accumulation of p53[25,26] and p53[25,26,53,54] mutant proteins in *Villin^Cre*; *p53^LSL-25,26/FL* and *Villin^Cre*; *p53^LSL-25,26,53,54/FL* mice, respectively, which

**Fig. 3 | p53 is required to induce revSC and FCC cells during intestinal regeneration. A** Schematic illustrating the generation of a query map (p53 KO: *Villin^Cre*; *p53^FL/−*) compared with the reference map (p53 WT: *Villin^Cre*; *p53^FL/+* and WT). (Created with BioRender.com). **B** UMAP plots depicting changes in cellular composition in mice with p53 WT or p53 KO intestinal epithelium in non-irradiated (d0) or two days (IR D2) after 12 Gy SBI. **C** Representative images from smFISH of *Clu* and *Ly6a* expression in mice with p53 WT or p53 KO intestinal epithelium in non-irradiated (NR) or 48 h and 60 h after irradiation. Scale bar = 50 μm. **D** Percentage of double *Clu* and *Ly6a*-positive cells quantified from **C**. The boxplot represents the IQR of the data, with the median indicated. The whiskers represent the highest and lowest values within 1.5 times the IQR. Each dot represents the average of 15 tissue areas quantified for each mouse (0 h: *n* = 3, 48 h: *n* = 3, and 60 h: *n* = 3). Statistical significance was assessed using a two-sided linear regression model for overall significance, and pairwise comparisons were made using the Bonferroni correction

method (*p* val = 0.0003159). **E** Schematic indicating that small intestinal organoids were treated with 4-hydroxytamoxifen (4-OHT) prior to irradiation (3 or 5 Gy) in control (DMSO) conditions or during p53-inhibition by Pifithrin-α (Pf-α) treatment. **F** Representative immunofluorescence images of organoids showing Clu and Clu progeny (Yellow) and Lgr5 (green) on the 5^th day of regeneration, grown in media containing either DMSO or Pf-α. White arrows indicate the absence of Clu+ cells and progeny in the crypts of irradiated organoids treated with Pf-α. (*n* = 3 experiments, 2 technical replicates). Scale bar = 100 and 50 μm. **G** Quantification of the percent of organoids displaying lineage-traced. Each dot represents an individual experiment (*n* = 3 experiments), with the number of organoids counted per experiment and per condition ranging from 17–106 (*n* = 17–106). The bars represent the mean percentage of organoids with more than 50% tdTOM-positive, and the error bars indicate the standard error of the mean. Statistical significance was calculated using a one-way ANOVA test followed by a Post-hoc Tukey's HSD test.

confirmed that p53 TAD domain mutant proteins were specifically expressed in GI epithelial cells[21] (Supplementary Fig. 5b).

To examine whether p53-dependent transcriptional programs were abrogated in mice with GI-specific p53 TAD mutant protein expression, we performed IHC on the intestine of unirradiated mice and mice 4 hours after 13.4 Gy SBI to detect the canonical p53 transcriptional target p21[27]. In mice that retained wild-type p53 (*p53^FL/+*), we observed induction of p21 after SBI. However, *Villin^Cre*; *p53^LSL-25,26/FL* and *Villin^Cre*; *p53^LSL-25,26,53,54/FL* mice failed to induce p21 after irradiation, similar to mice that lacked p53 (*Villin^Cre*; *p53^FL/FL*) (Fig. 5d, e). To evaluate a functional endpoint of the p53 transcriptional program, we also performed IHC for cleaved caspase-3 to quantify radiation-induced apoptosis in the GI epithelium, which is p53-dependent[28]. In mice that retained wild-type p53 (*p53^FL/+*), we observed a significant increase in the number of cleaved caspase-3 positive intestinal cells after SBI, but in mice expressing p53 TAD mutants (*Villin^Cre*; *p53^LSL−25,26/FL* or *Villin^Cre*; *p53^LSL-25,26,53,54/FL*), the number of cleaved caspase-3 positive intestinal cells was not significantly induced by irradiation and was similar to mice that lacked p53 (*Villin^Cre*; *p53^FL/FL*) (Supplementary Fig. 5c, d). Mice containing mutated p53 proteins in the GI epithelia were then treated with 13.4 Gy SBI and followed for radiation-induced GI syndrome. In contrast to mice that retained one wild-type allele of p53 in the GI epithelium (*Villin^Cre*; *p53^FL/+*), mice expressing p53 with mutations in TAD1 alone (*Villin^Cre*; *p53^LSL-25,26/FL*) or in both TAD1 and TAD2 (*Villin^Cre*; *p53^LSL-25,26,53,54/FL*) were sensitized to radiation-induced GI syndrome similar to mice that lacked p53 (*Villin^Cre*; *p53^FL/FL*) (Fig. 5f).

Moreover, we examined intestinal barrier function using the gold standard Dextran-FITC permeability assay[29]. Increased intestinal permeability is a sign of perturbed intestinal barrier function and has been associated with impaired proliferation and tissue regeneration[30,31]. We evaluated the levels of Dextran-FITC in the serum of *Villin^Cre*; *p53^FL/+* and *Villin^Cre*; *p53^FL/−* mice 5 days after 12 Gy to show that while in irradiated *Villin^Cre*; *p53^FL/+* mice the levels of Dextran-FITC in the serum were slightly higher than in unirradiated controls, *Villin^Cre*; *p53^FL/−* mice displayed significantly higher concentrations of Dextran-FITC in the serum compared to unirradiated mice and irradiated *Villin^Cre*; *p53^FL/+* mice (Fig. 5g). Collectively, our results demonstrate that the p53-dependent response to acute DNA damage is necessary to protect mice from radiation-induced GI syndrome by promoting the restoration of the mucosal barrier of the irradiated intestinal epithelium.

## Discussion

Our results from scRNA-seq, intestinal organoids, and genetically engineered mice harboring different p53 TAD mutations in the GI epithelium indicate that the p53 transcriptional response to acute DNA damage is required to produce revSCs and regenerate the epithelium after severe radiation damage. These findings provide insight into a long-standing question in radiation biology: how does the tumor

suppressor p53 suppress the development of radiation-induced GI syndrome?[9,32–34] Our results demonstrate that transient induction of p53-mediated signaling following irradiation is essential for the generation and reprogramming of Clu+ revSCs in the intestinal epithelium in vivo and intestinal organoids in vitro. However, prolonged activation of p53 in intestinal organoids post-irradiation via the continuous treatment of the Mdm2 inhibitor Nutlin-3 leads to decreased organoid budding, impaired clonogenicity, and loss of Clu+ revSCs as well as Lgr5+ CBCs. Collectively, these results reveal that the level of p53 protein is elegantly controlled through the p53-Mdm2 feedback loop to facilitate proper regeneration of the irradiated intestinal epithelium.

These findings do not contradict our previous study showing that the Super p53 mice, which harbor an extra copy of p53, are resistant to gastrointestinal acute radiation syndrome[9]. Although the extra copy of p53 in the Super p53 mice leads to a transient enhancement of p53-mediated signaling in the small intestines following irradiation, the activation of p53 protein also induces Mdm2 that subsequently degrades p53 protein. Indeed, given that the Super p53 mice exhibit higher induction of p53 downstream targets after irradiation, such as p21[35], it is conceivable that the expression of Mdm2 is also enhanced in the Super p53 mice following cellular stress to maintain the transient nature of p53 activation.

Collectively, we demonstrate that Clu+ revSCs contribute to a significant fraction of regenerating epithelial crypts in a mouse model of radiation-induced GI syndrome, and revSCs have also been previously shown to increase animal survival following gut damage[5]. Importantly, while acute inhibition of p53 impairs crypt regeneration via revSCs, prolonged p53 activation through inhibition of Mdm2 also suppresses the ability of revSCs to regenerate crypts following irradiation. These results define a key role of the p53-Mdm2 feedback loop in controlling the emergence of revSCs during intestinal regeneration following severe radiation injury.

## Methods

### Animal models

All procedures with mice were approved by the Institutional Animal Care and Use Committee (IACUC) of Duke University, by the IACUC of the University of Pennsylvania, and by the Canadian Council on Animal Care. All mouse strains used in this study have been described before, including *Villin^Cre*, *p53^FL/FL*, *p53^FL/−*, *p53^LSL-25,26*, *p53^LSL-25,26,53,54*, *Clu^EGFP*, *Clu^CreERT2*, *Lgr5^CreERT2*, *Lgr^-DTR-GFP*, Ai9, and Ai14 mice[5,21,36–42]. The *Villin^Cre* mice were originally obtained from the Jackson Laboratory and then bred at Duke University. The *p53^FL/FL* mice were originally provided by A. Berns (Netherlands Cancer Institute, Amsterdam, the Netherlands), and the *p53^LSL-25,26* and *p53^LSL-25,26,53,54* mice were originally provided by L. Attardi (Stanford University, Stanford, CA) and then bred at Duke University. The *Lgr5^CreERT2* were obtained from Jatin Roper (Duke University). The *Clu^EGFP* mice were originally obtained from Rockefeller University (GENSAT project) and then bred at the Lunenfeld-

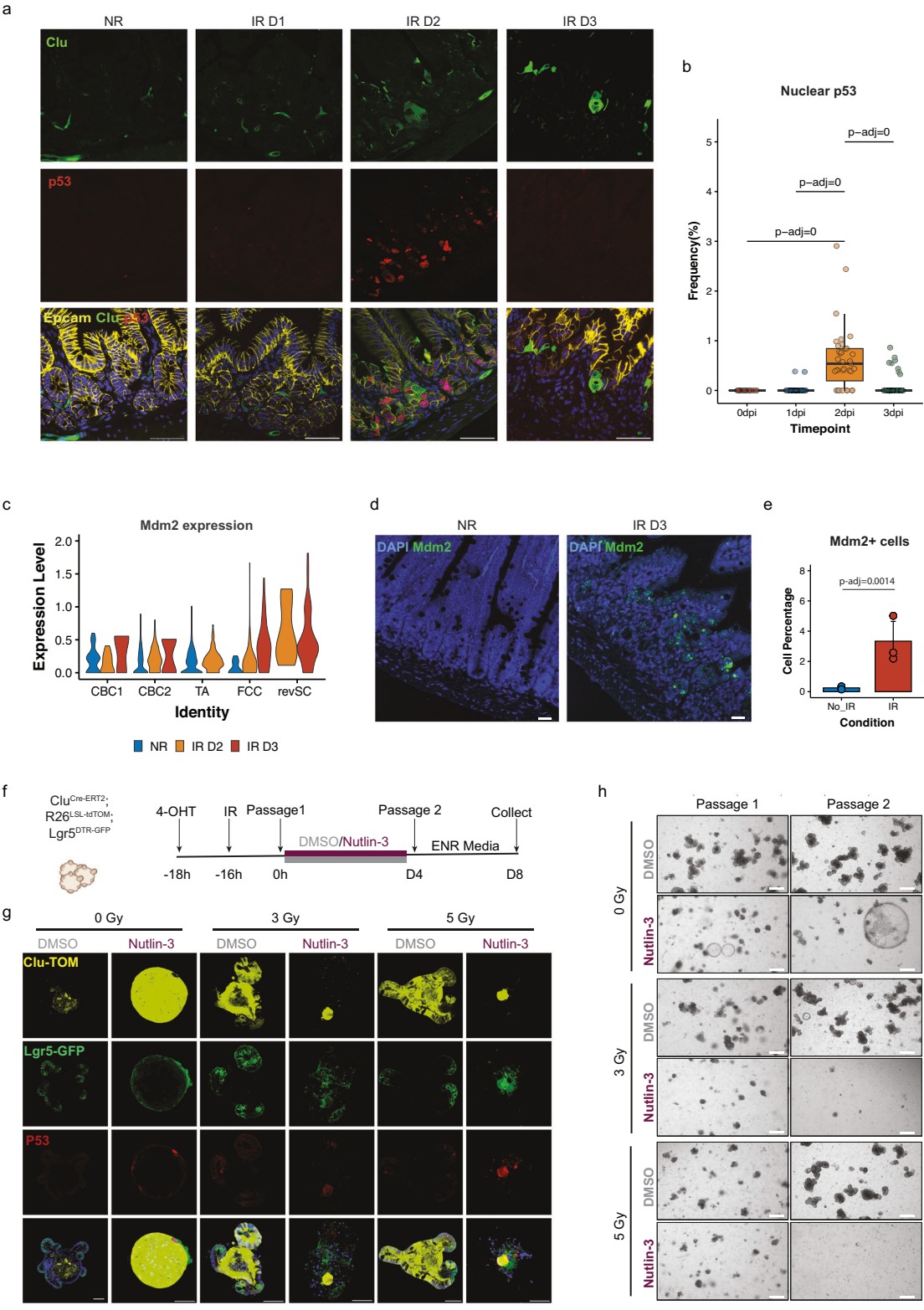

Tanenbaum Research Institute (Toronto, ON, Canada). The *Clu^CreERT2* mice were created and bred at the Lunenfeld-Tanenbaum Research Institute, and the methods of creating the mice have been described previously[5]. The *Lgr5^DTR-GFP* were provided by Frederic de Sauvage (Genentech, San Francisco, CA) and bred at the Lunenfeld-Tanenbaum Research Institute. The Ai9 and Ai14 mice were obtained from Jackson Laboratory and crossed with CreERT2 mice to generate reporter mice. Mice were at least 8 weeks old at the time of irradiation. Both sexes of mice were used, and littermates were used as controls for all experiments. The *Villin^Cre*, *p53^FL/FL*, *p53^FL/-*, *p53^LSL-25,26*, *p53^LSL-25,26,53,54* are on a mixed background that includes C57Bl/6 and 129 Sv/Ja. The *Lgr5^DTR-GFP* are C57Bl/6, and the *Clu^EGFP* are FVB/N-Crl:CD1 (ICR).

**Fig. 4 | Constitutive expression of p53 impairs intestinal tissue regeneration after irradiation. A** Immunofluorescence of Clu-GFP (green), P53 (red), and Epcam (yellow) on intestinal tissue sections 1, 2, and 3 days after 12 Gy SBI. Scale bar = 50 μm. **B** Quantification of the percentage of cells with nuclear P53 from **A**. Data from one experiment (3 mice/timepoint, $n = 3$) is shown, with each dot representing a single cross-section. The boxplot represents the interquartile range (IQR) of the data, with the median indicated. The whiskers extend from the box to the highest and lowest values within 1.5 times the IQR. Statistical significance was calculated using one-way ANOVA test followed by a Post-hoc Tukey's HSD test. **C** UMAP plots showing the *Mdm2* mRNA expression in revSC and FCC populations at day 0, 2, and 3 days after 12 Gy SBI. **D** Representative smFISH of Mdm2 transcript in mouse intestinal tissue sections from non-irradiated (NR) or 3 days (IR D3) after 12 Gy SBI. Scale bar = 50 μm. **E** Percentage of Mdm2 positive cells quantified in intestinal tissue sections represented in **D**. The bars represent the mean value of the data for each condition, with error bars indicating the standard deviation from the mean.

Each dot represents the average value of five tissue areas that were quantified for each mouse (non-irradiated: ($n = 2$) or irradiated: ($n = 3$)). Statistical significance was calculated using a two-sided $t$ test comparison. **F** Schematic indicating that small intestinal organoids were treated with 4-OHT to induce a lineage trace form Clu+ cells before irradiation (3 or 5 Gy), all prior to passaging. Organoid regeneration was tracked over a 4-day period, with growth under control (DMSO) or Nutlin-3 treatment. **G** Representative immunofluorescence images of Clu and Clu progeny (yellow), Lgr5 (green), and p53 (red), indicating that under continuous p53 activation (Nutlin-3), organoids show an inability to recover from IR conditions by day 4 of regeneration. ($n = 2$ experiments, 2 technical replicates/experiment). Scale bar = 50 μm. **H** Bright field images 1 and 2 passages after irradiation, indicating that organoids show an inability to recover from continuous p53 activation, with culture collapse evident after the initial treatments. ($n = 3$ experiments, 2 technical replicates/experiment). Scale bar = 500 μm.

## Intestinal organoids culture

Small intestinal organoids were cultured according to a previously described protocol established by Sato and Clevers[43]. Crypts were collected by incubating the small intestine in PBS containing 2 mM EDTA, followed by isolation using a 70 mm cell strainer. Crypts were seeded in growth factor reduced Matrigel (BD Biosciences) and grown in Advanced-DMEM/F12 (Life Technologies #12634028) supplemented with 2 mM GlutaMax (Life Technologies #35050061), 100 U/mL Penicillin/100 mg/mL Streptomycin (Life Technologies #15140122), N2 Supplement (Life Technologies #17502001), B-27 Supplement (Life Technologies #17504001), mouse recombinant Egf (Life Technologies #PMG8043), 100 ng/mL mouse recombinant Noggin (Peprotech #250-38) and 300 ng/mL human R-spondin1 (R&D Systems #4645-RS).

## Radiation experiments

**In vivo.** For radiation-induced GI syndrome and lineage tracing experiments, 8–10-week-old male and female mice were exposed to a single dose of SBI at Duke University. Jigs were used to hold unanesthetized mice and limit their movement. The head and the front limbs of mice were shielded by lead to prevent radiation-induced hematopoietic syndrome. SBI was performed with an XRAD 320 biological irradiator (Precision X-ray Inc., North Branford, CT). All mice were treated 50 cm from the radiation source, and a total physical dose of 10, 13.4, or 14 Gy using X-rays of 320 kV and 12.5 mA was delivered.

For the scRNA-seq experiment, 10–12-week-old male and female mice were exposed to a single dose of SBI at the University of Pennsylvania. Mice were immobilized with 2.5% isoflurane anesthesia with medical air as the carrier gas (VetEquip). The head of the mouse was placed in a facemask that allows the gas to be scavenged (Xerotec Inc.). To deliver whole abdominal irradiation, a variable collimator set at $20 \times 40$ mm (length $\times$ height) was used with the isocenter set at the center of the abdomen. A total physical dose of 12 Gy of X-rays using 220 kV and 13 mA was delivered to the mouse abdomen using the 3rd generation Small Animal Radiation Research Platform (SARRP by XStrahl).

**In vitro.** In vitro irradiation of organoids for lineage tracing experiments, *Clu^{CreERT2}; Rosa26^{LSL-tdTomato}; Lgr5^{DTR-GFP}* organoids were exposed to a single dose of 3 or 5 Gy using a GammaCell 40 irradiator to induce injury 16 hours before passage.

## Lineage tracing experiments

**In vivo.** To induce Cre-mediated gene recombination in vivo, *Clu^{Cre-ERT2}; Rosa26^{LSL-tdTomato}* or *Lgr5^{Cre-ERT2}; Rosa26^{LSL-tdTomato}* mice were administered 100 mg/kg tamoxifen (Sigma #T5648) via intraperitoneal injection at time points of 24 and 48 hours after sham-IR or SBI. Mice were euthanized 10 days post-SBI or sham-IR for tissue harvest. Lineage-traced crypts were quantified manually by counting crypts with ≥70% tdTomato signal. The percentage of lineage tracing events per slide

was determined by dividing the total number of positive tdTomato crypts by the total number of crypts present. The images were quantified by two observers (S.S. and L.D.), who were blinded to the treatment. A total of 10 representative 10× images from proximal to distal were taken for each slide. The number of *Clu^{Cre-ERT2}; Rosa26^{LSL-tdTomato}* mice quantified were $n = 3$ 0 Gy, $n = 4$ 10 Gy, and $n = 4$ 14 Gy. The number of *Lgr5^{Cre-ERT2}; Rosa26^{LSL-tdTomat-}* mice quantified were $n = 3$ 0 Gy, and $n = 1$ 14 Gy. The number of mice analyzed was determined by the number of mice that did not develop the radiation-induced GI syndrome prior to 10 days post-IR.

**In vitro.** For in vitro lineage tracing experiments, *Clu^{-CreERT2}; Rosa26^{LSL-tdTomato}; Lgr5^{DTR-GFP}* organoids were treated with 4-hydroxytamoxifen (4-OHT; Sigma #H6278; 5 μg/ml) to induce a lineage trace and two hours later were irradiated as described above. At the time of passaging, organoids were grown under mock conditions, with ENR (Egf/Noggin/R-spondin) media containing DMSO, or with either Nutlin3a (Sigma #SML0580; 10 μM) to activate the p53 pathway or Cyclic Pifithrin-α hydrobromide (PF-α; Sigma #P4236; 20 μM) to inhibit the p53 pathway. Following the desired regeneration period, lineage-traced crypts were quantified manually by counting the number of Lgr5-GFP+ regions that displayed ≥70% tdTomato signal.

## Tissue histology and staining

**Immunohistochemistry (IHC).** Unirradiated and irradiated mice were sacrificed at indicated time points, and the middle 10–12 cm length (jejunum) of the small intestine was isolated and flushed with cold DPBS (Gibco). The tissue was then divided into 2 cm segments for 10% formalin fixation for 16–24 hours. Prior to embedding, segments were bundled using 3 M Micropore Surgical Tape (3 M) for cross-sectioning.

For immunohistochemistry (IHC), deparaffinized slides underwent 3% hydrogen peroxide treatment, antigen retrieval with either citrate- or Tris-based solution (Vector Laboratories), and blocking with 5% goat serum. Slides were incubated with primary antibody against p21 (1:100; Ref: ab188224; Clone: ERP18021; Abcam), cleaved-caspase-3 (1:500, Ref: #9664, Clone: 5A1E; Cell Signaling Technology), or p53 (1:200, Ref: NCL-L-p53-CM5p; Leica Biosystems). For chromogenic detection, the secondary antibodies used include Goat anti-Rabbit IgG (H + L) Cross-Adsorbed Secondary Antibody, Biotin (31822; Invitrogen). The VECTASTAIN Elite ABC system (Vector Laboratories) was applied, and 3,3'-diaminobenzidine (DAB) was used as chromogen. Slides were imaged using a Leica DM2000 LED phase contrast microscope. Cells positive for cleaved-caspase-3 signals were quantified by individual crypt, and the examiner was blinded to the genotype and the treatment of the samples. Briefly, the examiner was provided with a ×20 deidentified histology slide of intestinal crypts. The examiner would then determine if any intact crypts were present. Intact crypts were defined as any crypts with a clearly demarcated base lined with Paneth cells that also had at least 10 additional cell nuclei on either side

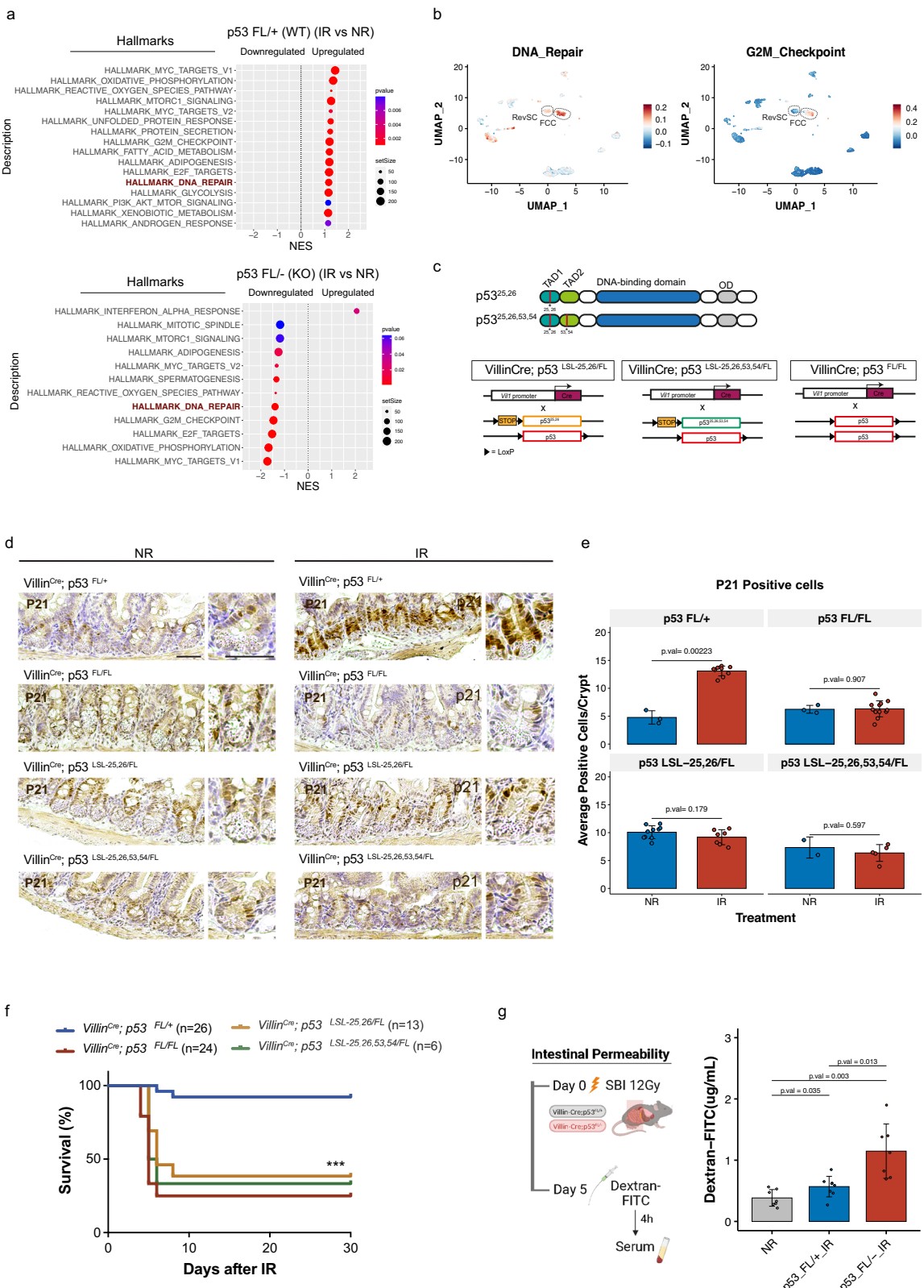

of the crypt column beyond position 4, which is the generally accepted position of intestinal CBCs. If an intact crypt was identified, the number of positively staining cells in that individual crypt was recorded. Data was compiled using Microsoft Excel.

**Immunofluorescence (IF).** The small intestines were harvested, flushed with ice-cold PBS, cut into smaller fragments, and fixed overnight

in 10% buffered formalin. For cryopreservation, fixed intestines were next transferred to 30% sucrose in PBS overnight at 4 °C and stored in OCT at −80 °C. OCT sections (16-µm thick) were then permeabilized in 0.5% Triton (Sigma) and 0.2% Tween (Sigma #P1379) in PBS for 30 min, blocked in 2.5% BSA for 1 h and mounted in Mowiol mounting Media with DAPI (Invitrogen). Images were taken at ×10 for crypt quantification, and 2.5× for representative full images using a Leica fluorescent

**Fig. 5 | p53 TAD domains are required to prevent radiation-induced GI syndrome. A** Gene set enrichment analysis (GSEA) showing top upregulated or downregulated gene sets two days after irradiation (IR) in p53 WT or p53 KO intestinal epithelial cells. *P* values were calculated using the empirical Bayes moderated *t* test from the Limma package. **B** UMAPs showing the expression of the DNA Repair and G2M Checkpoint gene signatures across epithelial cell clusters in intestinal tissues of p53 WT mice 3 days after 12 SBI. **C** Schematic of p53 protein domains showing the locations of the transactivation domains (TAD) mutation residues (upper) and schematic of p53 alleles of the *Villin^Cre* mice used in this study (lower). TAD1 and TAD2: transactivation domains 1 and 2. OD: oligomerization domain. **D** Representative images of immunohistochemistry staining of p21 protein on intestinal tissue sections from p53 WT and p53 mutants non-irradiated (No IR) or 4 hours after 13.3 Gy SBI. Scale bars = 50 μm. **E** Quantification of p21 staining from **D**. p21-positive cells within the +4 to +10 region of the crypt were counted from

intestinal tissue sections from p53$^{FL/+}$ (NR: *n* = 3, IR: *n* = 9), p53$^{FL/FL}$(NR: *n* = 3, IR: *n* = 12), p53$^{LSL.25,26/FL}$ (NR: *n* = 10, IR: *n* = 7) and p53$^{LSL.25,26,53,54/FL}$ (NR: *n* = 2, IR: *n* = 5) mice. For each tissue section, 10–15 crypts were quantified. The bars represent the mean value of the data for each treatment within each genotype, with error bars indicating the standard deviation from the mean. Statistical significance was calculated using a two-sided *t* test. **F** Kaplan–Meier curves of p53 WT (p53$^{FL/+}$) or p53 mutants (p53$^{FL/FL}$, p53$^{LSL.25,26/FL}$, and p53$^{LSL.25,26,53,54/FL}$) mice following 13.4 Gy of SBI (*n* = 69). *P* val < 0.0001 is calculated using a log-rank test. **G** The schematic illustrates the evaluation of intestinal permeability in p53 WT and KO after IR. (Created with BioRender.com). Dextran-FITC concentration in the blood of p53 WT (NR: *n* = 6, IR: *n* = 8) and p53 KO (NR: *n* = 1, IR: *n* = 7) mice at day 5 after 12 Gy of SBI. The bars represent the mean value of the Dextran-FITC concentration for each genotype, with error bars indicating the standard deviation (SD) from the mean. Statistical significance was calculated using a two-sided *t* test.

microscope. For p53, Epcam, and Clu-GFP staining, slides were incubated with antibodies to p53 protein (CM5) (Leica Biosystems), EpCAM (CD326) (G8.8) (118201; BioLegend) and GFP (Abcam #ab13970; 1:1000). To quantify p53 activation in the intestinal epithelia, all images were acquired at ×63 on a Nikon Ti2 confocal microscope system. Images were processed in batch format in CellProfiler[44]. DAPI counterstaining was used to create nuclear masks following measurement of p53 signal intensity in each nucleus, which was used to calculate frequency of cells showing p53 activation per image across all treatments.

For IF in intestinal organoids, organoids were grown on glass coverslips for 3–5 days after passaging. At the desired timepoint, organoids were fixed in 4% PFA (Electron Microscopy Sciences #15710) for 30 minutes, followed by permeabilization and blocking for 1 hour at room temperature in PBS with 0.5% Triton-X-100 (Sigma #X100-500ML)/0.2% Tween-20 (Sigma #P1379)/2% BSA (Sigma #A7906). Coverslips were incubated overnight at 4 °C in primary antibodies targeted against p53 protein (CM5) (Leica Biosystems; 1:300) and GFP (Abcam #ab13970; 1:1000). Secondary DyLight antibodies (Fisher Scientific; 1:1000) were used. Sections were counterstained with DAPI (Sigma-Aldrich) for 30 minutes before mounting. Images were acquired using a Nikon Ti2 inverted confocal microscope platform.

**RNA in situ hybridization.** RNA in situ hybridization (ISH) was performed with RNAscope technology (Advanced Cell Diagnostics). The probes to detect *Ly6a*, *Clu*, *mKi67,* and *Mdm2* mRNA were MmLy6a-427571, MmCLuC3-427891-C3 and Mmki67C2-416771 and MmMdm2C2-447641-C2, respectively. For chromogenic assay, ISH was conducted using RNAscope 2.5 HD Assay-BROWN as per the manufacturer's protocol. For fluorescent assay, RNAscope Multiplex Fluorescent V2 Assay was used. Either the Opal Fluorophore reagent packs (Opal 520, Opal 570, Opal 690) (Akoya Biosciences) or the tyramide signal amplification (TSA) fluorophores (TSA Fluorescein, TSA Cyanine 3, TSA Cyanine 5) (Akoya Biosciences) were used for detection. Images were acquired using a Nikon Ti2 inverted confocal microscope platform.

**RNA ISH Image quantification.** mKi67, Ly6a, and Clu positive cells from RNA ISH tissue sections were quantified using QuPath v0.3.2 software[45]. All cells were first detected based on DAPI staining using the *Cell detection* command. For each staining, a classifier was generated using the *Train object classifier* command. The three classifiers were then combined and run across all images. For each timepoint between 3 and 6 mice were harvested, and total of 15–20 images were quantified for each intestinal swiss roll.

**Intestinal barrier permeability assay**
Villin$^{Cre}$;p53$^{FL/+}$ and Villin$^{Cre}$;p53$^{FL/-}$ mice were subjected to 12 Gy of SBI. At day five post-IR, non-irradiated controls and irradiated mice were

first subjected to 4 h hours of fasting and then were administered with 100 μl of 80 mg/mL 4 KDa FITC-dextran (Sigma #D4-1G) by oral gavage. 1 hour post-gavage blood was collected, followed by serum extraction. Fluorescence in serum samples was measured using an Infinite Pro2000 plate reader.

**Intestinal Crypt isolation and single-cell suspension protocol**
Small intestines were removed, and ~15 cm of the jejunum was isolated and washed with ice-cold 1× HBSS (Cytiva, #SH3058801), opened longitudinally, and divided into three pieces of roughly equal length. To obtain the epithelial cells, intestinal fragments were incubated in epithelial cell solution (10 mM EDTA [Invitrogen, #15575020]-HBSS, 5 mM HEPES [Invitrogen, #15630080], 100 U/ml Pen-Strep [Invitrogen, #15140122], and 2% FBS [Life Technologies, #26140079]) for 30 min on ice with continuous gentle shaking. Tissue fragments were then transferred into fresh ice-cold HBSS. Vigorous shaking was used to isolate the epithelial fraction. Intestinal crypts were recovered by filtering the tissue suspension through a 70-μm mesh-cell strainer. Three consecutive epithelial fractions were collected, pooled, and centrifuged for 10 minutes at 500 × *g*. Epithelial cell pellets were resuspended in pre-warmed Trypsin-EDTA (Corning), and incubated 5 min at 37 °C. Single cells were obtained by continuous pipetting for 15–20 minutes at room temperature (RT). Trypsin was quenched by the addition of 100% FBS and cells were washed twice with HBSS and finally filtered through a 40-μm mesh-cell-strainer.

**Flow cytometry cell sorting and 10X single-cell RNA-sequencing processing**
Intestinal single cells were resuspended in fluorescence-activated cell sorting (FACS) buffer (Advanced-DMEM/F12 with 5% FBS) containing 0.1 μg/mL of DAPI. For each sample, 1 × 10$^5$ alive single cells were sorted in FACS buffer. After sorting, cells were washed twice with ice-cold PBS and counted again using an automated cell counter. 1 × 10$^4$ cells for each sample were subjected to 10X Genomics single-cell isolation using the Chromium Next GEM Single-Cell 3' v3.1 Kit (10x Genomics) following the manufacturer's recommendations. Single-Cell 3' Gene Expression libraries were sequenced using an Illumina NovaSeq 6000 platform.

**Single-cell RNA-Seq data processing and integration**
The scRNA-seq data was processed using the Cell Ranger pipeline (10x Genomics) to demultiplex the FASTQ reads, align them to the mm10 Mouse genome, and generate gene-barcode expression matrices. Quality control of these expression matrices was conducted using the Seurat R package to retain cells with greater than 500 unique UMIs, more than 250 detected genes, a logarithmic ratio of detected genes per unique UMI above 0.8, and a proportion of counts in mitochondrial genes less than 10%. Additionally, only genes with non-zero counts in more than ten cells were kept, and doublet removal was performed

using scDblFinder. Data was then normalized and scale using the NormalizeData and ScaleData functions from Seurat Packages. Finally, samples were integrated using the Harmony package. Epithelial cells were first identified, and subset based on the expression of the epithelial gene marker *Epcam*. To test how single-cell clusters identified in wild-type intestinal epithelia were altered upon p53 mutation, single cell RNA-seq data set generated from p53 mutant intestinal epithelia was first processed as above by using the same parameters that were applied to process the wild-type data set. Next, FindTransferAnchors function implemented in Seurat was employed to compare the two datasets, where pre-calculated wild-type clustering was used as a reference map to infer single-cell clusters in the mutant data set. MapQuery function was then used to generate a query map of mutant single-cell clusters and plot the results. Identification of specific gene markers for all clusters was calculated using Seurat's FindAllMarkers default settings using a a Wilcoxon Rank Sum test. Epithelial cell clusters were then annotated based on the expression of canonical stem cell and differentiation markers previously described[5,46–48]. Enterocytes were further classified based on the expression of previously described villus zonation gene signatures[10]. To visualize the expression of specific genes across cell clusters and conditions FeaturePlot, VlnPlot, and pheatmap functions were used. To interrogate the average expression of gene signatures at single-cell level the AddModuleScore function was used. Differential gene expression was conducted using the default voom and limma-trend pipeline. Subsequent gene set enrichment analyses based on all differentially expressed genes were conducted with the Hallmark databases using the ClusterProfiler R package. For Supplementary Fig. 2b, c, raw data was downloaded from Cheung et al, and processed following the code provided in the publication. FeaturePlot function was used to query the expression of revSC and p53 target genes into the UMAP.

## Statistical analysis and reproducibility

Statistical analyses were performed using the R statistical software (version 4.3.1). To assess differences between groups, we conducted one-way ANOVA tests followed by Tukey's HSD post hoc tests using the 'stats' and 'multcomp' packages in R. Significance was set at a $p$ value of <0.05, and Bonferroni correction was applied for multiple comparisons. For assessing the differences in double positice Clu and Ly6a cell numbers between genotypes across time, linear modeling was conducted to examine the relationships of these two variables. The analysis was performed using linear models in R, facilitated by the 'lm' function from the 'stats' package. When comparing only two groups, we employed t-tests. The significance level for these tests was also set at a $p$ value of <0.05. For Kaplan–Meier curves, a Gehan-Breslow-Wilcoxon test was applied.

## Reporting summary

Further information on research design is available in the Nature Portfolio Reporting Summary linked to this article.

## Data availability

The single-cell RNA-sequencing data generated in this study have been deposited in the SRA database under accession code SRR22589001. The single-cell RNA-sequencing data from Ayyaz et al. Nature 2019 and the single-cell RNA-sequencing data used in Supplementary Fig. 2 are available from Gene Expression Omnibus (GEO) with accession codes GSE123516 and GSE152376, respectively. Source data are provided in this paper.

## Code availability

The code used to visualize the scRNA-sequencing data is available in GitHub repository.

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

## Acknowledgements

We thank members of our laboratories for helpful feedback, Anton Berns from the Netherlands Cancer Institute for providing the p53^FL/FL mice, and the Cell and Animal Radiation Core Facility (RRID: SCR_022377) at the University of Pennsylvania Perelman School of Medicine. The funding sources for this manuscript are National Institutes of Health 2U19AI067798 (C.-L.L. and D.G.K.), National Institutes of Health U19AI067773 (C.-L.L.), National Institutes of Health 2R35CA197616 (D.G.K.), National Institutes of Health 1P01CA257904 (C.K., A.J.M., C.-L.L., and D.G.K.), Duke University School of Medicine Whitehead Scholar Award (C.-L.L.), The Mark Foundation for Cancer Research (A.J.M. and C.M.M.).

## Author contributions

H.-C.K., A.A., C.M.M., A.J.M., J.W., C.-L.L., and D.G.K. conceptualized the project; H.-C.K., A.A., C.M.M., M.F., I.V., A.R.D., D.N.B., L.D., S.S., S.H., R.G., L.L., and Y.M. performed experiments; H.-C.K., A.A., C.M.M., M.F., and L.D.A. developed methodology and reagents; C.K., A.J.M., J.W., C.-L.L., and D.G.K. acquired funding; C.K., A.J.M., J.W., C.-L.L., and D.G.K. supervised the project; C.-L.L. and D.G.K. wrote the original draft. All authors reviewed and edited the final manuscript; J.W., C.-L.L., and D.G.K. contributed equally to this manuscript.

## Competing interests

D.G.K. is a co-founder of and stockholder in XRad Therapeutics, which is developing radiosensitizers. D.G.K. is a member of the scientific advisory board and owns stock in Lumicell Inc., a company commercializing intraoperative imaging technology. None of these affiliations represents a conflict of interest with respect to the work described in this manuscript. D.G.K. is a coinventor on a patent for a handheld imaging device and is a coinventor on a patent for radiosensitizers. XRad Therapeutics, Merck, Bristol Myers Squibb, and Varian Medical Systems provided research support to D.G.K., but this did not support the research described in this manuscript. C.-L.L. reports research support from Rythera Therapeutics; however, this does not present a conflict of interest with the content of this manuscript. The remaining authors declare no competing interests.
