## [Peer Review File · Nature Communications]

p53 promotes revival stem cells in the regenerating intestine after severe radiation injuryEditorial Note: Parts of this Peer Review File have been redacted as indicated to maintain the confidentiality of unpublished data.

REVIEWER COMMENTS

Reviewer #1 (Remarks to the Author):

The paper by Morral et. al. has investigated the mechanism of p53 mediated stem cell regeneration after severe radiation injury. They build on decades of work in the field, led by the senior corresponding author on the role of p53 in the intestinal radiation response. It is known that p53 dosage affects the ability of the intestines to regenerate after radiation, but the mechanisms of how this occurs are not well understood. The authors revisit the question of how p53 regulates the radiation response by using some new tools that have been developed since that last investigations, namely scRNA seq and the use of NTAD p53 mutants that can parse our canonical and non-canonical effects.

The authors single cell RNA sequencing on cells from small intestine of mouse with functional p53 or deleted p53 after radiation and organoids represent a tremendous advance in the field, and will likely serve as a reference point for others doing similar studies.

The authors build on previous work (Ayyaz, et al Nature 2019) identifying Clu+ revival stem cells as a key population to regenerate the intestine after radiation injury. The authors demonstrate that transient p53 expression in Clu+ revSC cells are critical for intestinal regeneration after radiation injury.

Overall, this is a tremendous piece of work using multiple novel mouse models and a robust scRNA seq analysis to find deeper understanding for the role of p53 in the radiation response of the intestinal tract. Major Comments:

1. The scRNA seq analysis appears very robust and the authors should be commended. I would highly encourage the authors to pledge to release their code for the analysis to assist other scientists wishing to replicate their important analyses.
2. The authors have focused on Clu+ rev SCs, which are a relatively new cell population. Did the authors find any role for the classical +4 cells and could the authors comment on the role of p53 in these two important cells types.
3. The authors demonstrate that Nutlin-3 in organoids impaired the response to radiation. However, this results appear to contradict their own previous findings using "Super P53" mice that have reduced radiation injury when p53 is overexpressed (Kirsch et al Science 2010). A more likely explanation are off target effects of Nutlin rather than specific effects on Mdm2. The authors should address this experimentally or through at least through discussion.
4. Withers-Elkind Crypt assays are still considered the gold standard in evaluating crypt regeneration, and would strengthen the manuscript. However, if these are not done, the authors should comment on the reasons for not performing these assays (technical, mouse background, etc...)

Minor comments:

5. Fig 3; authors showing p53 induction at day 1 and day 2 post radiation, however the classical p53 pathway initiated immediately following radiation, did the authors look at early expression of p53 like 4-6 hours post radiation?

6. Fig 4; authors showing p53 mediated p21 induction by IHC at 4hrs post radiation however in Fig 3 showing p53 induction at day 2 post radiation. Also, it is not clear from the images why there is more p21 in p53LSL-25,26/FL. p21 expression could be shown by western blot.

Reviewer #2 (Remarks to the Author):

Comments for Author

In this manuscript, Morral et al. found that p53 have a pivotal role to control cell reprogramming after radiation-induced GI damages. The function of p53 during the process is to induce revival stem cells which are known as reserve stem cells, damage-induced quiescent cell type in mouse intestine. The results of their single cell RNA-seq data also demonstrated that p53 expression was transiently increased in revival stem cells after irradiation (IR) and down-regulated by Mdm2. The strength of the paper is the identification of the negative feedback loop of p53 and Mdm2 during their regeneration step of intestinal epithelium following radiation injury. However, there are some critical points need to be addressed to strengthen their study.

Major comments

1. My main comment is about the molecular mechanism of p53 how to induce revival stem cells upon radiation injury. The manuscript at this moment is still descriptive. Recently, there are several findings concerning reprogramming/plasticity in intestinal epithelium, and it has been reported that YAP signaling is a crucial role for that. Is there any the relationship between p53 and YAP signaling in the revSC during regenerating steps?

Furthermore, the authors also found the pathways related with the DNA damage response, G2/M checkpoint, and DNA repair in Fig.4A. The authors should address whether those events are observed in FCC and revSC during reprogramming steps. Those studies are likely to shed light on the molecular function of p53 in development of FCC and revSC.

2. Line 77: The description of scRNA-seq method is insufficient. I do not know why their RNA-seq results represented the data of only epithelial cells. The method they used, EDTA based crypts isolation, always contaminates non-epithelial tissues e.g. mesenchymal cells, neuron, etc. Do the authors use the antibody against epithelial cells such as EpCAM for FACS?

3. Line 102: The authors found Ly6a mRNA expression is observed in both cell types of Clu positive and Clu negative in intestinal epithelium after IR. Because Fig. S1H contains critical information, it should included in the main figure. Despite the modest quantity of red dots, I still noticed the signalis of Clu expression in the Ly6a+ cells. That's why their conclusion, especially the part of Ly6a expression in Clu negative cells, might lead misunderstanding. To acquire a clear result, the authors need to conduct that double staining of Ly6d and Clu together with E-cadherin with reference to the following paper, Itzkovitz et al. Nat Cell Bio. 2011.

4. Line 130-132: Ayyaz et al. (Nature 2019) previously shown endogenous Clu expression in intestine in wild-type mice that is comparable to Fig. 2C (IR, 48h, p53FL/+) in this paper. However, Clu expression was not observed at all in Fig. 2C No IR, d0) control in this paper. Is the difference due to p53 heterozygous mutant?

5. Line 154, Fig.2F: The Experimental procedures for counting Lgr5+/tdTomato+ cells are unknown. In the image of organoid treated with Pf-a and 5 Gy, I could see many yellow cells. Isn't this result an expansion of Clu+ lineage cells?

6. Line 168: One of the key findings in this paper is the negative feedback of p53 and Mdm2. Therefore, the authors need to show Mdm2 expression in both level of mRNA and protein in intestinal tissues.

7. Fig.3F and G: The authors used Nutlin-3 to activate p53 pathway. However, I am wondering why the wild-type organoid are survived in the present of Nutlin-3 in control (0 Gy). Nutlin-3 is widely used in various organoid works, especially to eliminate wild-type organoid when cancer organoid is established. The concentration of Nutlin-3 is not specified, but I would want the authors to explain why the wild type of organoid survives.

8. Fig. 2E and 3E: It is still unclear if those organoid trials with small molecule compounds on p53 activity, Pf-a, Nutlin-3, were successful. Real time q-PCR and Western blotting analyses for p53, Mdm2, and p21 are necessary in those organoid experiments.

9. Fig. 4C related with Fig.3A-D: It has been known that p53 undergoes post-transcriptional modifications such as phosphorylation and acetylation, which leads to its stability and activation when DNA damage occurs. Concomitantly, p53 induces the target genes such as p21, Mdm2 and so on. These processes usually evolve within a few hours. In fact, the authors demonstrated p53 activation following DNA damage by observing increased p21 expression in 4 hours in Fig4C. However, it took 2 days to activate p53 after IR in Fig. 3A-C. The authors need to explain those temporal discrepancy.

Minor comments:

1. Line 71-72, 210: In this paper, the genotypes of p53 are written in a mixed manner. Especially, what is the difference between Villin-Cre; p53FL/- and Villin-Cre;p53FL/FL? The authors need to explain more detail about those genotypes.

2. Fig. S1B: Lack information of cell cluster 9. What cell types does cluster 9 represent?

3. Line 140: What does it mean of LD20/10 and LD50/10?

4. Line 150 : Authors used two different genotype of Lgr5 mouse, Lgr5DTR-GFP and Lgr5-GFP-iDTR (in Methods line 546). What is Lgr5-GFP-iDTR mouse? I know Sauvage et al. generated Lgr5-DTR-eGFP (Tian et al. Nature 2011), but not Lgr5-GFP-iDTR. Authors need to explain details if they generated inducible Lgr5-DTR mouse newly.

5. Line 541: What is the genetic background all mice and crosses used in the study? This is very important both for the current study, and for comparison with earlier published studies.

Reviewer #3 (Remarks to the Author):

In this manuscript, the authors described the involvement of p53 in regenerative intestine after radiation injury. While the manuscript provides a potentially important protective role of p53, the data are preliminary and descriptive. There are three major concerns.

1. The mechanisms underlying the protective role of p53 is not clear. The authors provided evidence that p53-dependent transcription is important, but p53 controls a vast transcriptional program that drives many functions such as anti-oxidative, autophagy, and other metabolic and cell survival activities. It is not clear which function(s) of p53 is involved.
2. The authors only focused on the p53 involvement in the regeneration after lethal IR dosages. What happens when the IR dosages are lower? This will help to understand the mechanism of such protective role.
3. In the context of physiological and clinical relevance, p53 is commonly mutated in intestinal cancers. the authors should extend their analysis to p53 mutant mouse models such as R175H, R248W knock-in mice.

We thank all Reviewers for their insightful and thoughtful comments and suggestions. We are grateful for the overall positive comments, including comments by Reviewer 1, who stated, “this is a tremendous piece of work using multiple novel mouse models and a robust sc-RNA seq analysis to find deeper understanding for the role of p53 in the radiation response of the intestinal tract”. We also appreciate the general critique about the temporal dynamics of p53 induction and the importance of investigating the underlying molecular mechanism, which has motivated us to perform additional experiments and improve the conceptual advance of our study. Finally, we also appreciate Reviewers 1 and 2 for their valuable minor comments on experimental details that helped us to improve the rigor and reproducibility of our study. Below, we describe the major changes to the revised manuscript and then provide point-by-point responses to individual Reviewer comments. New data generated during this revision is included in this rebuttal letter and in the new manuscript as detailed below.

A. MAJOR CHANGES

Temporal dynamics of p53 activation, molecular mechanism, and phenotypic characterization of p53 KO mice

Temporal Dynamics of p53 Activation:

We thank Reviewers 1 and 2 for highlighting the importance of understanding the temporal dynamics of p53 activation in the context of radiation-induced acute gastrointestinal (GI) injury. In our initial manuscript, we primarily focused on the role of p53 in tissue regeneration at later time points (days 2-3 post-damage). However, in response to your feedback, we have expanded our investigation to include both early and late time points, resulting in a more comprehensive study of p53 dynamics during intestinal regeneration (**Supplementary Figure 3A-B**).

Our findings reveal a novel two-wave pattern of p53 activation. The first wave occurs very early after tissue damage, as shown in our time course by a slight increase of p53 protein at 4h after IR that rapidly returned to undetectable levels. *Stewart-Ornstein et al., 2021* (PMID: 33563973) showed that in intestinal tissues, p53 has a peak of induction around 2h after IR and is rapidly degraded, returning to almost undetectable levels around 4-5 hours after IR. These results could explain the low number of p53-positive cells per crypt found in our intestinal tissues 4h after IR. The second wave of p53 activation takes place around day 2-3 after IR, coinciding with the emergence of Clu+ revSCs (**Figure 4A-B**). The temporal overlap of this second wave of p53 expression with revSC appearance, along with the absence of revSCs in p53 knockout (p53 KO) mice, indicates a critical role of p53 in mediating the regeneration of irradiated intestinal epithelium through revSCs. This novel function of p53, beyond its canonical role in DNA damage response, could provide insights into how p53 confers tissue radioprotection independently of apoptosis, as previously observed (*Kirsch et al., Science* 2010; PMID: 20019247). Thus, our revised study sheds light on this novel mechanism by which p53 protects mice from radiation-induced acute GI injury.

Molecular Mechanism Underlying p53-Mediated Tissue Regeneration:

We appreciate the Reviewers' interest in understanding the molecular mechanisms underlying p53-mediated tissue regeneration. In the original manuscript, we conducted functional experiments in intestinal organoids to demonstrate that p53 activation must be transient to induce Clu+ revSCs and that Mdm2 plays a crucial regulatory role in this process. In the revised

manuscript, we have included Mdm2 staining in tissue sections to further support these findings (**Figure 4D-E**).

Importantly, in the original manuscript, we explored the expression of some of the p53-downstream targets, such as *Cdkn1a*, *Bax*, and *Phlda3*, and showed their enrichment in the revSC population. In the revised manuscript we now examine the expression of the entire p53 signature and showed its specific enrichment in the revSCs (**Figure 2A**). Following Reviewer 2's suggestion to investigate the potential interaction between p53 and YAP signaling pathways in revSC induction, we analyzed our scRNA-seq data to show that the YAP gene signature is enriched in revSC during tissue regeneration. This observation aligns with our previous report indicating that YAP signaling promotes revSC emergence (Ayyaz *et al.*, *Nature* 2019; PMID: 31019301). Furthermore, using a published single-cell RNA sequencing dataset, we found that hyperactivation of YAP in intestinal epithelial cells induces the expression of p53 and its downstream targets, suggesting a potential role of p53 signaling in mediating YAP-induced revSC induction. We have not included these new findings in the revised manuscript (only in this rebuttal letter) as they represent a new direction of investigation requiring further experimental work and time to fully investigate, which is beyond the scope of this study.

Phenotype Associated with p53 KO Mice:

Reviewer 1 raised concerns about the phenotype associated with lower survival in mice lacking p53 after IR and suggested investigating the proportion of regenerating crypts as a surrogate for tissue regeneration. In line with Hendry *et al.* (PMID: 9291357), we found no difference in the number of regenerating crypts based on BrdU incorporation at day 4 after 12 Gy. The BrdU assay merely measures the number of cells entering the S phase but does not provide insights into the structural and functional state of the epithelial barrier. Thus, we performed an intestinal permeability assay to directly measure intestinal barrier dysfunction and found that p53 KO mice exhibited increased barrier permeability compared to WT mice after 12 Gy (**Figure 5G**). These results provide novel insight into the differences in survival between the two genotypes. We have included these findings in the revised manuscript, providing further evidence that p53 plays a crucial role in mediating the regeneration of irradiated intestinal epithelium and protecting mice from radiation-induced GI injury.

B. REVIEWER COMMENTS

Reviewer #1 (Remarks to the Author):

The paper by Morral *et. al.* has investigated the mechanism of p53-mediated stem cell regeneration after severe radiation injury. They build on decades of work in the field, led by the senior corresponding author on the role of p53 in the intestinal radiation response. It is known that p53 dosage affects the ability of the intestines to regenerate after radiation, but the mechanisms of how this occurs are not well understood. The authors revisit the question of how p53 regulates the radiation response by using some new tools that have been developed since that last investigations, namely scRNA seq and the use of NTAD p53 mutants that can parse our canonical and non-canonical effects.

The authors single cell RNA sequencing on cells from the small intestine of mice with functional p53 or deleted p53 after radiation and organoids represent a tremendous advance in the field, and will likely serve as a reference point for others doing similar studies.

The authors build on previous work (Ayyaz, et al Nature 2019) identifying Clu+ revival stem cells as a key population to regenerate the intestine after radiation injury. The authors demonstrate that transient p53 expression in Clu+ revSC cells is critical for intestinal regeneration after radiation injury.

Overall, this is a tremendous piece of work using multiple novel mouse models and a robust scRNA seq analysis to find a deeper understanding of the role of p53 in the radiation response of the intestinal tract.

Major Comments:

1) The scRNA seq analysis appears very robust and the authors should be commended. I would highly encourage the authors to pledge to release their code for the analysis to assist other scientists wishing to replicate their important analyses.

We are very thankful for this positive comment. The code for the computational analysis is deposited in our GitHub repository: https://github.com/claramorral/P53_revSC
Raw sequencing data has already been deposited in GEO.

2) The authors have focused on Clu+ revSCs, which are a relatively new cell population. Did the authors find any role for the classical +4 cells and could the authors comment on the role of p53 in these two important cell types?

We thank the reviewer for this insightful question. The +4 cells -often referred to as Label retaining cells (LRCs)- have been previously suggested to be an alternative stem cell population located at the position +4 from the intestinal crypt base that expresses a specific gene signature that includes genes such as *Bmi1*, *Tert*, *Hopx* and *Lrig1* (PMID: 18536716; PMID: 21173232; PMID: 22075725; PMID: 22464327). We have interrogated the expression of the LRC gene signature in our data set, but we did not detect significant enrichment of these genes in any of the identified epithelial cell clusters (see **Rebuttal Figure 1A** and **1B** and **Supplementary Figure 1E** of the revised manuscript). We also assessed the individual expression of these genes across our epithelial clusters both during homeostasis (Day 0), and regeneration (Day 2 and Day 3). While *Bmi1* and *Lrig1* exhibited scattered expression in various clusters, *Tert1* expression was minimal and *Hopx* showed low expression primarily in the crypt base columnar cells (CBCs), the transit amplifying (TA), and some differentiated cells from the secretory lineage (**Figure 1C** of this rebuttal, not included in the manuscript). In conclusion, our analysis suggests that these genes do not define a distinct population of reserve stem cells in our dataset. This observation aligns with previous findings, including those from the Clevers group, which showed similar gene expression patterns in Lgr5 CBC cells and no specific enrichment outside the stem cell zone (Muñoz et al., 2012; PMID: 22692129). Additionally, Arshad et al. (PMID: 31019301) demonstrated that the LRC gene markers were expressed in various clusters with no discernible pattern, further supporting our results (see Extended Data Fig. 3b of Arshad et al., *Nature* 2019). We are very thankful to the reviewer for motivating us to look at these cells, and we hope these additional analyses address your concerns adequately.

Rebuttal Figure 1: LRC gene signature expression in intestinal epithelial cells in homeostasis and after irradiation. (A) UMAP showing all epithelial clusters identified. **(B)** Heatmap representing the average expression of the LRC gene signature across the epithelial clusters. **(C)** Individual UMAPs showing the expression of LRC genes in homeostasis (No IR) and during regeneration (IR Day 2 and, Day 3) in P53 WT epithelial intestinal cells.

3) The authors demonstrate that Nutlin-3 in organoids impaired the response to radiation. However, these results appear to contradict their own previous findings using “Super P53” mice that have reduced radiation injury when p53 is overexpressed (Kirsch et al Science 2010). A more likely explanation are off target effects of Nutlin rather than specific effects on Mdm2. The authors should address this experimentally or through at least through discussion.

We appreciate this comment from the Reviewer. We agree this is a very important point that we have discussed in the Discussion of the revised manuscript as the following:

“Our results demonstrate that transient induction of p53-mediated signaling following irradiation is essential for the generation of Clu+ revSCs in the intestinal epithelium *in vivo* and intestinal organoids *in vitro*. However, prolonged activation of p53 in intestinal organoids post-irradiation via the continuous treatment of the Mdm2 inhibitor Nutlin3 leads to decreased organoid budding, impaired clonogenicity, and loss of Clu+ revSCs as well as Lgr5+ intestinal stem cells (ISCs). Collectively, these results reveal that the level of p53 protein is finely controlled through the p53-Mdm2 feedback loop to facilitate proper regeneration of the irradiated intestinal epithelium.

These findings do not contradict our previous study showing that the Super p53 mice, which harbor an extra copy of p53, are resistant to gastrointestinal acute radiation syndrome (*Kirsch et al., Science* 2010; PMID: 20019247). Although the extra copy of p53 in the Super p53 mice leads to a transient enhancement of p53-mediated signaling in the small intestines following irradiation, the activation of p53 protein also induces Mdm2 that subsequently degrades p53 protein. Indeed, given that the Super p53 mice exhibit higher induction of p53 downstream targets after irradiation such as p21, it is plausible that the expression of Mdm2 is also enhanced in the Super p53 mice following cellular stress to maintain the transient nature of p53 activation.”

We appreciate the reviewer's astute observation and hope that this explanation clarifies the relationship between our findings with Nutlin-3 and the previous "Super p53" mouse model results. This discussion is now included in the revised manuscript for further context and clarity.

4) Withers-Elkind Crypt assays are still considered the gold standard in evaluating crypt regeneration and would strengthen the manuscript. However, if these are not done, the authors should comment on the reasons for not performing these assays (technical, mouse background, etc...)

We thank the reviewer for this suggestion. Notably, a paper from Chris Potten's group showed that there was no difference in the levels of crypt survival in the small intestine between p53+/, p53+/- and p53-/- mice 4 days after 14 Gy. In addition, they observed that the number of crypts 3 days after 14 Gy was even higher in p53-/- mice compared to p53+/+ mice based on H&E staining (*Hendry et al., 1997*; PMID: 9291357). We also performed a similar experiment by quantifying the number of surviving crypts containing ten or more BrdU-labelled cells per circumference in the small intestines 4 days after 13.4 Gy SBI. Our results mirrored those reported by Hendry et al., showing no significant difference between Villin-Cre; p53 FL/+ and Villin-Cre; p53 FL/FL littermates. (**Rebuttal Figure 2A and B**). However, these findings were not included in the revised manuscript as they did not provide additional information to support the manuscript's primary message.

Rebuttal Figure 2: BrdU staining in regenerating intestine. (A) Representative images of BrdU IHC in regenerating intestines 96h after 13.4Gy SBI in P53 FL/+ and P53 FL/FL mice. (B) Quantification of BrdU-positive crypts. A crypt is considered positive if it has > 10 BrdU+ cells. (Each dot represents a quantified image, 3-4 images/mouse. For p53 FL/+ = 4 mice and p53 FL/FL=8 mice).

Although we did not observe significant differences in crypt survival using the BrdU assay, prompted by the reviewer's suggestion, we investigated further the reduced survival observed in p53 FL/FL mice after SBI. Increased intestinal permeability is a sign of perturbed intestinal barrier function and has been associated with impaired proliferation and tissue regeneration. Thus, we interrogated intestinal barrier function using the gold standard Dextran-FITC permeability assay (<https://en.bio-protocol.org/en/bpdetail?id=1289&type=0>). We evaluated the levels of Dextran-FITC in the blood of Villin-Cre; p53 FL/+ and Villin-Cre; p53 FL/- mice. We chose day 5 after IR because we know that at this time point, the intestinal barrier is being restored unless there is a regeneration defect. Our results showed that while in irradiated Villin-Cre; p53 FL/+ mice the levels of Dextran in the serum were slightly higher than in unirradiated controls, Villin-Cre; p53 FL/- mice displayed significantly higher concentrations of Dextran-FITC in the serum compared to unirradiated mice and irradiated Villin-Cre; p53 FL/+ mice (**Rebuttal Figure 3** and **Figure 5G** of the manuscript). We have included these results in the new manuscript as they provide additional evidence supporting our model, which suggests that the absence of p53 in the intestinal epithelium sensitizes mice to acute gastrointestinal radiation syndrome due to impaired intestinal regeneration.

Rebuttal Figure 3: P53 FL/- mice show reduced proliferation and increased intestinal barrier permeability after IR. (A) Schematic representation for the Dextran-FITC assay. (B) Dextran-FITC concentration in the serum of p53 FL/+ and p53 FL/- mice NR or 5 days after IR.

Minor comments:

5) Fig 3; authors show p53 induction at day 1 and day 2 post radiation, however, the classical p53 pathway initiated immediately following radiation, did the authors look at early expression of p53 like 4-6 hours post radiation?

We appreciate the reviewer bringing up this point, as we did not provide evidence of p53 induction at early time points in our original manuscript. To address this comment, we performed p53 IHC staining in the same tissue sections used to evaluate the activation of the p53 downstream targets p21 and cleaved caspase 3. We found that in Villin-Cre; p53 FL/+ mice the number of nuclear p53+ cells per crypt was significantly higher 4 hours post-12 Gy compared to unirradiated controls, whereas we could not detect p53 protein expression in Villin-Cre; p53 FL/FL mice with and without irradiation as expected. (**Rebuttal Figure 4** and **Supplementary Figure 3A** and **B** of the revised manuscript).

One limitation of this experiment is that the tissues were collected 4 hours after irradiation. In *Stewart-Ornstein et al., 2021* (PMID: 33563973) the authors explored the temporal dynamics of p53 induction after IR and showed that in intestinal tissues p53 has a peak of induction around 2h after IR and is rapidly degraded returning to almost undetectable levels around 4-5 hours after IR. These results could explain the low number of p53-positive cells per crypt found in our intestinal tissues 4h after IR. Importantly, this study also shows that the transient induction of p53 correlates with higher radioprotection compared to other tissues -such as the spleen- in which p53 activation is sustained over time. Indeed, sustained p53 activation using an Mdm2 inhibitor is sufficient to sensitize tumor cells to death. Although this study focuses on the classical p53-mediated DNA damage response at early time points after damage, we believe that it supports very well the notion that transient activation of p53 is required to confer radioprotection.

In the revised manuscript, we added p53 quantification data at a 4h time point in the p53 time course provided in the original manuscript (**Supplementary Figure 3A-B**). The updated time course clearly shows two waves of p53 activation after IR. An initial wave at early time points (probably even higher at 2h) and a second wave of activation between 2-3 days after IR that promotes the emergence of revival stem cells to support tissue regeneration. In addition, we included a working model in the revised manuscript to better explain the implications of the findings in our study (**Supplementary Figure 3E**).

Rebuttal Figure 4. P53 expression dynamics after irradiation in intestinal tissues. (A) Representative IHC of P53 in tissue sections from Villin-Cre; p53 FL/+ mice across several time points after IR. **(B)** Quantification of A.

6) Fig 4; authors showing p53 mediated p21 induction by IHC at 4hrs post radiation however in Fig 3 showing p53 induction at day 2 post radiation. Also, it is not clear from the images why there is more p21 in p53LSL-25,26/FL. p21 expression could be shown by western blot.

We appreciate the reviewer raising this concern. As shown in **Rebuttal Figure 4**, the new results in our revised manuscript showed that the expression of p53 protein in intestinal epithelial cells was significantly induced in two waves following severe radiation injury (**Supplementary Figure 3A-B**). Thus, the induction of p21 protein by IHC 4 hours post-irradiation is the consequence of the first wave of p53 protein activation in response to acute DNA damage.

As the Reviewer pointed out, it is intriguing that the baseline level of p21+ epithelial cells was higher in VillinCre; p53 LSL-25,26/FL mice. It has been demonstrated that p53 25,26 protein still retains some transactivation activity and is more stable than wild-type p53 protein because of disruption of the residues required for Mdm2 binding (PMID: 21565614). Indeed, a paper from co-author Laura Attardi's lab showed that the baseline expression of a subset of p53 transcriptional targets such as p21 was significantly higher in p53 25,26/- mouse embryos compared to p53 +/- embryos (PMID: 31178404). Therefore, the increase in p21+ crypts in Villin-Cre; p53 LSL-25,26/FL mice can be explained by the stabilization of p53 25,26 protein that has impaired, but not completely abrogated, transactivation activity to induce p21.

However, our data showed that irradiation does not further induce the number of p21+ crypt cells in Villin-Cre; p53 LSL-25,26/FL mice. These results are consistent with p53 IHC showing that p53 25,26 protein was expressed in almost every intestinal epithelial cell without irradiation and could not be further increased 4 hours after 13.4 Gy (**Supplementary Figure 4B**).

Reviewer #2 (Remarks to the Author):

Comments for Author

In this manuscript, Morral et al. found that p53 has a pivotal role to control cell reprogramming after radiation-induced GI damages. The function of p53 during the process is to induce revival stem cells which are known as reserve stem cells, damage-induced quiescent cell type in mouse intestine. The results of their single cell RNA-seq data also demonstrated that p53 expression was transiently increased in revival stem cells after irradiation (IR) and down-regulated by Mdm2. The strength of the paper is the identification of the negative feedback loop of p53 and Mdm2 during their regeneration step of intestinal epithelium following radiation injury. However, there are some critical points need to be addressed to strengthen their study.

Major comments

1) My main comment is about the molecular mechanism of p53 how to induce revival stem cells upon radiation injury. The manuscript at this moment is still descriptive. Recently, there are several findings concerning reprogramming/plasticity in intestinal epithelium, and it has been reported that YAP signaling is a crucial role for that. Is there any the relationship between p53 and YAP signaling in the revSC during regenerating steps?

We appreciate this comment from the Reviewer and agree that our manuscript could be improved by giving more attention to the molecular mechanism. Specifically, the reviewer's comment is about the possible relationship between p53 and YAP signaling in the revSC as YAP signaling plays a crucial role during regeneration. The original manuscript shows that the p53 transcriptional program is specifically activated in revSCs. Following the reviewer's suggestion, we interrogated the expression of downstream YAP target genes and found that these genes are also enriched in the revSC population after IR (**Rebuttal Figure 5**). These results support the observations in *Ayyaz et al., 2019* (PMID: 31019301) showing that the YAP gene signature was present exclusively in the revSC cluster. Importantly, the authors demonstrate that depletion of *Yap1* in the intestinal epithelium abrogated the emergence of *CLU+* cells. Accordingly, knockout of *Last1* and *Lats2*, which activates YAP, was sufficient to induce ectopic *Clu* cells in the homeostatic intestine.

YAP gene signature

Rebuttal Figure 5. YAP gene signature is enriched in revSCs in the regenerating intestine. UMAPPs showing the expression of YAP gene signature across epithelial cell clusters during homeostasis (NR) and after IR (day 2 and day 3).

Although we have not addressed the exact mechanism by which p53 and YAP signaling pathways interact, a body of literature indicates the existence of crosstalk between these two signaling pathways. For example, *Bai et al., 2013* (PMID: 23760493) showed that overexpression of YAP resulted in increased expression of p53 and that both the TEAD and WW domains were required for YAP-mediated p53 function. To further explore the potential relationship between YAP and p53 signaling, we revisited a single-cell RNA sequencing data set published by *Cheung et al., 2020* (PMID: 32730753). In this study, the authors deleted the LATS1/2 kinases in the mouse intestinal stem cells to induce YAP hyperactivation and subsequently performed scRNA seq. Using this data set, we interrogated the expression of revSCs gene markers in addition to well-known p53 downstream targets. We found that hyperactivation of YAP (Lats1/2 KO) induced the upregulation of both sets of genes in intestinal stem cells, supporting the notion that YAP signaling can promote p53 activation and cooperate to induce revSCs (**Rebuttal Figure 6**). We haven't included this data in the new manuscript, but we believe that these results reinforce the connection between these two signaling pathways and how they cooperate to control tissue regeneration. The exact mechanisms of how YAP signaling activates p53 would require a substantial amount of experimental work and time, and we think that these experiments are beyond the scope of this manuscript.

Rebuttal Figure 6. YAP activation induces the expression of revSC and P53-target genes in the intestinal epithelium. UMAP plots showing the expression of revSC genes and P53 downstream targets in intestinal epithelium from WT mice or mice that lack LATS1 and 2.

Furthermore, the authors also found the pathways related with the DNA damage response, G2/M checkpoint, and DNA repair in Fig.4A. The authors should address whether those events are observed in FCC and revSC during reprogramming steps. Those studies are likely to shed light on the molecular function of p53 in development of FCC and revSC.

Our original manuscript showed that DNA repair and G2M checkpoint were among the top enriched gene sets in intestinal epithelial cells from p53 FL/+ mice after IR. Accordingly, these genes were downregulated in p53 FL/-, suggesting the lack of p53 impairs the activation of DNA damage response-related genes. Following the reviewer's suggestion, we have explored in which cell types these pathways are activated during regeneration. In the revised manuscript we show that DNA repair genes are enriched predominantly in the FCCs but also in the revSCs at day 3 after IR, which aligns with our previous results showing that p53 downstream target genes are enriched in these two populations. Interestingly, the G2M Checkpoint gene set seems to be specifically activated in the FCC cell cluster but not in the revSCs. We reasoned that since the FCC represents an undifferentiated high proliferative population, activation of the G2/M DNA damage checkpoint possibly through replicative stress prevents entrance into mitosis with damaged DNA to prevent death in these cells (**Rebuttal Figure 7** and **Figure 5B** of the reviewed manuscript).

Rebuttal Figure 7. DNA repair and G2M checkpoint genes are activated in revSCs and FCC after irradiation. UMAP plots showing the expression DNA Repair and G2M Checkpoint gene signature across epithelial clusters 3 days after irradiation.

2) Line 77: The description of scRNA-seq method is insufficient. I do not know why their RNA-seq results represented the data of only epithelial cells. The method they used, EDTA based crypts isolation, always contaminates non-epithelial tissues e.g. mesenchymal cells, neuron, etc. Do the authors use the antibody against epithelial cells such as EpCAM for FACS?

Thank you for this suggestion. In the new manuscript, we have now included in **Supplementary Figure 1A** a schematic representation of how epithelial cells were selected during the sc-RNA sequencing analysis. In brief, we selected cells that expressed the epithelial marker *Epcam* and subset them from other cells expressing immune and stromal-related genes. Since we enriched intestinal crypts during the harvesting procedure, the epithelial population represented >80% of total cells obtained for single-cell RNA seq analysis. These cells were next re-clustered and used for downstream analyses.

3) Line 102: The authors found Ly6a mRNA expression is observed in both cell types of Clu positive and Clu negative in intestinal epithelium after IR. Because Fig. S1H contains critical information, it should included in the main figure. Despite the modest quantity of red dots, I still noticed the signals of Clu expression in the Ly6a+ cells. That's why their conclusion, especially the part of Ly6a expression in Clu negative cells, might lead misunderstanding. To acquire a clear result, the authors need to conduct that double staining of Ly6d and Clu together with E-cadherin with reference to the following paper, Itzkovitz et al. Nat Cell Bio. 2011.

We appreciate the reviewer's comment concerning Ly6a mRNA expression in both Clu-positive and Clu-negative cell types within the intestinal epithelium after irradiation, and we agree that we should clarify these observations in the manuscript. Following the reviewer's suggestion we have included in the main **Figure 1E and F** (see also **Rebuttal Figure 8**) new images of Ly6a and Clu ISH staining in intestinal tissues 3 days after IR that clearly show the presence of Ly6a+Clu- cells (white arrowheads). To provide stronger evidence, we have also quantified the overlap between Ly6a and Clu in these images and demonstrated that while up to 70% of Clu+ cells also express Ly6a, Clu is only expressed in 25% of Ly6a+ cells, reinforcing the presence of Ly6a+Clu- cells. These results align with our scRNA seq data showing two distinct populations of undifferentiated cells that emerge after irradiation, Ly6a+; Clu- proliferating cells and Ly6a+; Clu+ quiescent revSCs. We appreciate the reviewer's suggestion to include staining with an epithelial marker such as E-Cadherin, but we believe that the ISH and single-cell RNA seq data together sufficiently characterize these populations as epithelial cells. Moreover, Villin-Cre is known to be GI epithelial cell-specific and the changes we observe in the Ly6a+ populations in the single-cell RNA seq data from Villin-Cre mice that delete p53 in the GI epithelial cells further support this interpretation.

Rebuttal Figure 8. Clu and Ly6a ISH in intestinal tissues 3 days after IR. (A) Representative images of Ly6a (yellow) and Clu (red) ISH staining in intestinal tissues 3 days after IR. White arrowhead denoted the presence of Ly6a+Clu- cells. **(B)** Percentage of double Ly6a and Clu positive cells. Orange bar represents the percentage of cells that are both Ly6a+ and Clu+ within the Clu cell population. The red bar represents the percentage of cells that are both Clu+ and Ly6a+ within the Ly6a cell population.

4) Line 130-132: Ayyaz et al. (Nature 2019) previously shown endogenous Clu expression in intestine in wild-type mice that is comparable to Fig, 2C (IR, 48h, p53FL/+) in this paper. However, Clu expression was not observed at all in Fig. 2C No IR, d0) control in this paper. Is the difference due to p53 heterozygous mutant?

We appreciate the reviewer's observation regarding Clu expression in our study compared to Ayyaz et al., 2019 (PMID: 31019301), and the potential influence of p53 heterozygosity on these differences. We previously showed that in the undamaged intestine, Clu+ cells are extremely rare (<0.1%; Ayyaz et al., 2019; Figure ED4e) and directly differentiate/convert into the main epithelial lineages, including Lgr5+ ISCs, which was shown using a lineage-tracing model. The rarity of these cells at homeostasis makes their quantification challenging. To further address this concern, we reexamined the single-cell RNA sequencing data from Ayyaz et al., 2019, and the results are presented in **Rebuttal Figure 9**. Our analysis revealed that we could only detect Clu+ cells in the wild type (WT) irradiated intestines, aligning with what was previously reported in Ayyaz et al., 2019, and reported in the current manuscript with p53FL/+ intestinal epithelial cells.

Ayyaz et al., Nature 2019

Rebuttal Figure 9. Clu+ cells are absent in homeostatic intestines but emerge after irradiation. UMAPs showing the expression of Clu mRNA in epithelial cells from homeostatic intestines (normal) and after irradiation (irradiated).

5) Line 154, Fig.2F: The Experimental procedures for counting Lgr5+/tdTomato+ cells are unknown. In the image of organoid treated with Pf-a and 5 Gy, I could see many yellow cells. Isn't this result an expansion of Clu+ lineage cells?

We thank the reviewer for raising these two points; we agree they should be clarified. In response to the first point, we have now included a more detailed explanation of how the Lgr5+/tdTomato cells were counted in the RNA ISH quantification section of the Materials and Methods.

Regarding the second comment, Clu can be induced in response to various forms of stress. These cells appearing in the lumen of the organoids likely represent cells in which Clu was

induced in cells that are sloughed off into the lumen as the organoid grows. Importantly, these data show that they were not functional revival stem cells that convert to Lgr5+ISCs to repopulate the organoids. If functional revSCs had been produced, this would be seen in the form of tracing from Lgr5+tdTomato+ cells throughout the body of the organoid. We have now included this explanation in the legend of Figure 3F of the revised manuscript.

6) Line 168: One of the key findings in this paper is the negative feedback of p53 and Mdm2. Therefore, the authors need to show Mdm2 expression in both level of mRNA and protein in intestinal tissues.

We appreciate the reviewer's comment that underscores the importance of providing comprehensive evidence for the mechanism discussed in our manuscript. Following the reviewer's suggestion, we have performed in situ hybridization (ISH) of Mdm2 mRNA in intestinal tissues during homeostasis and at day 3 post-irradiation. Our results from the ISH demonstrate that Mdm2 is not expressed in the normal intestinal epithelium but is markedly induced in irradiated tissues from Villin-Cre-p53 FL/+ mice. It's worth noting that we chose ISH as the method of choice due to the unavailability of suitable antibodies for immunohistochemistry or immunofluorescence in tissue sections. We have now included these results in the main manuscript (**Figure 4D and E and Rebuttal Figure 10**).

Rebuttal Figure 10. Mdm2 is induced in intestinal epithelial cells after irradiation. (A) Representative ISH of Mdm2 mRNA in intestinal tissues from Villin-Cre; P53FL/+ mice non-irradiated (NR) or 3 days after IR (IR D3). (B) Percentage of cells Mdm2+ in the intestinal epithelium from non-irradiated and 3 days after irradiation.

7) Fig.3F and G: The authors used Nutlin-3 to activate p53 pathway. However, I am wondering why the wild-type organoid are survived in the present of Nutlin-3 in control (0 Gy). Nutlin-3 is widely used in various organoid works, especially to eliminate wild-type organoid when cancer organoid is established. The concentration of Nutlin-3 is not specified, but I would want the authors to explain why the wild type of organoid survives.

Thank you, we appreciate this comment, and we think it is important to clarify it. Nutlin-3 is often used to select for p53 mutant organoids, as those that harbor WT p53 will ultimately not survive under conditions where p53 is being continuously activated. However, we expect that the organoids we show survive based on the duration of time spent in Nutlin-3+ media. The selection

process can take several days and varies by tissue type. For example, p53 WT lung organoids can take up to 11 days of treatment with Nutlin-3 before viability is at 0%, while organoids derived from biliary tract carcinoma still showed above 0% viability at 6 days of treatment with 10 μ M doses of Nutlin-3 (PMID: 33846488; PMID: 31018139). Here we have treated intestinal organoids for 4 days, and while there is a noticeable difference in the viability of organoids, some still survive at this time point.

The concentration of Nutlin-3 used for the lineage tracing experiments in organoids is 10 μ M as described in the material and methods section of the original manuscript.

8) Fig. 2E and 3E: It is still unclear if those organoid trials with small molecule compounds on p53 activity, Pf-a, Nutlin-3, were successful. Real time q-PCR and Western blotting analyses for p53, Mdm2, and p21 are necessary in those organoid experiments.

We are thankful to the reviewer for asking about this control experiment. To assess the effectiveness of the Mdm2 inhibitor Nutlin-3 we quantified the percentage of cells with nuclear p53 in intestinal organoids that were either mock-treated or treated with Nutlin-3 after exposure to 0, 3, or 5 Gy of irradiation. Our results from these experiments clearly demonstrate that p53 accumulates in cells from organoids treated with Nutlin-3. Furthermore, we observed a gradual increase in the number of p53+ cells with increasing irradiation dose, providing strong evidence of the efficacy of Nutlin-3 to inhibit the negative regulation of p53. We have included these results in the new version of the manuscript (see **Supplementary Figure 3D** and also **Rebuttal Figure 11**).

Rebuttal Figure 11. Stabilization of P53 in organoids treated with Mdm2 inhibitor. Quantification of the percent of cells in organoids with nuclear p53 signal following irradiation and given 5-day regeneration period +/- Nutlin-3.

Following reviewer question we performed RT-qPCR on Pf-a treated organoids. Our results showed that Pf-a treatment of intestinal organoids after IR, did not restore the expression of canonical p53 targets such as *Phlda3*, *Bax*, *Mdmd2* or *P21* to baseline levels (**Rebuttal Figure 12**). These results are in line with Zhu, J.et al., 2020 (PMID: 31974452), showing that in a model of P53 reactivation using Nutlin3, Pf-a was able to prevent the expression of a subset of p53 downstream targets only when cells were pre-treated 12h with the inhibitor but not when cells were treated at the same time p53 was induced (see Figure 2 of et al Zhu, J.et al., 2020). Similarly,

in our experimental design, organoids were first IR and then treated with the inhibitor and harvested 24h after for RT-qPCR analysis.

Importantly, our manuscript strongly suggests that p53 also regulates the transcription of FCC and revSC-associated genes (*Clu*, *Anxa2*, *Anxa1*, *S100a6*), which we know are important for tissue regeneration. Thus, we also interrogated the expression of these genes in our Pf-a treated organoids by RT-qPCR and found a partial restoration of these genes in the Pf-a treated condition (**Rebuttal Figure 12**). This result explains the reduced lineage tracing and impaired organoid regeneration observed in Pf-a treated organoids (**Figure 3** of the manuscript). We don't have a mechanistic explanation of why Pf-a inhibitor is not very efficient in preventing p53 canonical activity in our organoid model, but importantly we show that it reduces the revSC gene signature which is essential to drive regeneration.

Rebuttal Figure 12. Effect of Pfithrin-a on the transcriptional response of irradiated intestinal organoids. Expression of revSC and p53 associated genes in control (NT) or Pfithrin-a treated organoids non-irradiated (0Gy) or 24h after 5Gy irradiation.

[REDACTED]

Rebuttal Figure 13. [REDACTED]

9. Fig. 4C related with Fig.3A-D: It has been known that p53 undergoes post-transcriptional modifications such as phosphorylation and acetylation, which leads to its stability and activation when DNA damage occurs. Concomitantly, p53 induces the target genes such as p21, Mdm2 and so on. These processes usually evolve within a few hours. In fact, the authors demonstrated p53 activation following DNA damage by observing increased p21 expression in 4 hours in Fig4C. However, it took 2 days to activate p53 after IR in Fig. 3A-C. The authors need to explain those temporal discrepancy.

This is an excellent point and has also been asked by Reviewer 1 (see comment 5 by Reviewer 1). This is the response that we provided above to address this concern:

We appreciate the reviewer bringing up this point, as we did not provide evidence of p53 induction at early time points in our original manuscript. To address this comment, we performed p53 IHC staining in the same tissue sections used to evaluate the activation of the p53 downstream targets p21 and cleaved caspase 3. We found that in Villin-Cre; p53 FL/+ mice the number of nuclear p53+ cells per crypt was significantly higher 4 hours post-12 Gy compared to unirradiated controls, whereas we could not detect p53 protein expression in Villin-Cre; p53 FL/FL mice with and without irradiation as expected. (**Rebuttal Figure 4** and **Supplementary Figure 3A** and **B** of the revised manuscript).

One limitation of this experiment is that the tissues were collected 4 hours after irradiation. In *Stewart-Ornstein et al., 2021* (PMID: 33563973) the authors explored the temporal dynamics of p53 induction after IR and showed that in intestinal tissues p53 has a peak of induction around 2h after IR and is rapidly degraded returning to almost undetectable levels around 4-5 hours after IR. These results could explain the low number of p53-positive cells per crypt found in our intestinal tissues 4h after IR. Importantly, this study also shows that the transient induction of p53 correlates with higher radioprotection compared to other tissues -such as the spleen- in which p53 activation is sustained over time. Indeed, sustained p53 activation using an Mdm2 inhibitor is sufficient to sensitize tumor cells to death. Although this study focuses on the classical p53-mediated DNA damage response at early time points after damage, we believe that it supports very well the notion that transient activation of p53 is required to confer radioprotection.

In the revised manuscript, we added p53 quantification data at a 4h time point in the p53 time course provided in the original manuscript (**Supplementary Figure 3A-B**). The updated time course clearly shows two waves of p53 activation after IR. An initial wave at early time points (probably even higher at 2h) and a second wave of activation between 2-3 days after IR that promotes the emergence of revival stem cells to support tissue regeneration. In addition, we included a working model in the revised manuscript to better explain the implications of the findings in our study (**Supplementary Figure 3E**).

Minor comments:

1. Line 71-72, 210: In this paper, the genotypes of p53 are written in a mixed manner. Especially, what is the difference between Villin-Cre; p53FL/- and Villin-Cre;p53FL/FL? The authors need to explain more detail about those genotypes.

We thank the Reviewer for bringing up this point, and we apologize for not having clarified the difference between the two genotypes. The difference between the Villin-Cre; p53 FL/FL

and Villin-Cre; p53 FL/- models is that in Villin-Cre; p53 FL/FL mice the cells that are NOT GI epithelial cells (ie not expressing Villin or a descendant of a Villin-expressing cell) will express 2 wild-type copies of p53 while in Villin-Cre; p53 FL/- mice these non-GI epithelial cells express only 1 copy of p53. The rationale for using Villin-Cre; p53 FL/- mice is that only 1 allele of p53 needs to be recombined by Cre, so this increases the likelihood that p53 is fully deleted in all GI epithelial cells. This is the model originally used in the *Kirsch et al., Science 2010* (PMID: 20019247). Our subsequent studies showed that VillinCre; p53 FL/- and VillinCre; p53 FL/FL mice showed the same radiosensitization phenotype following SBI. Therefore, Villin-Cre; p53 FL/FL mice were used in certain experiments in the manuscript for the ease of breeding and genotyping and are an appropriate littermate control when the parents have an LSL-p53 point mutant rather than a wild type or null allele.

2. Fig. S1B: Lack information of cell cluster 9. What cell types does cluster 9 represent?

We thank the Reviewer for this comment. We missed this cluster in our previous annotation. Cells from cluster 9 are characterized by the expression of differentiation markers such as *Krt20* and *Fabp1* but lack mature absorptive enterocyte markers, such as *Alpi* or *Emp1*, so it likely reflects an enterocyte precursor. We have now added this label.

3. Line 140: What does it mean of LD20/10 and LD50/10?

LD20/10 and LD50/10 are the radiation doses that cause radiation-induced acute GI syndrome within 10 days post-IR in 20% and 50% of mice, respectively.

4. Line 150 : Authors used two different genotype of Lgr5 mouse, Lgr5DTR-GFP and Lgr5-GFP-iDTR (in Methods line 546). What is Lgr5-GFP-DTR mouse? I know Sauvage et al. generated Lgr5-DTR-eGFP (Tian et al. Nature 2011), but not Lgr5-GFP-iDTR. Authors need to explain details if they generated inducible Lgr5-DTR mouse newly.

We thank the reviewer for looking carefully at our manuscript and identifying this mistake. The mouse model used for this study is the Lgr5-DTR-GFP generated in the Sauvage laboratory and described in *Tian et al., 2011* (PMID: 21927002). We have now corrected the genotype throughout the text and figures.

5. Line 541: What is the genetic background all mice and crosses used in the study? This is very important both for the current study, and for comparison with earlier published studies.

We agree with the reviewer that providing the genetic background is important for the reproducibility of our study. We have provided this information in the Animal Models section of the new manuscript.

Reviewer #3 (Remarks to the Author):

In this manuscript, the authors described the involvement of p53 in regenerative intestine after radiation injury. While the manuscript provides a potentially important protective role of p53, the data are preliminary and descriptive. There are three major concerns.

1. The mechanisms underlying the protective role of p53 is not clear. The authors provided evidence that p53-dependent transcription is important, but p53 controls a vast transcriptional program that drives many functions such as anti-oxidative, autophagy, and other metabolic and cell survival activities. It is not clear which function (s) of p53 is involved.

We agree with the Reviewer that investigating the mechanism by which p53 protects mice from radiation-induced GI syndrome is an important question. Indeed, this is why we performed experiments using mice harboring various p53 TAD mutations. While p53 25,26,53,54 mutations inhibit the transcription of all p53 target genes, p53 25,26 mutations only impair the transcription of a subset of p53 targets that regulate the response to acute DNA damage. Notably, our data showed that mice harboring p53 25,26 mutations in intestinal epithelial cells (Villin-Cre; LSL-p53 25,26/FL) were sensitized to the radiation-induced GI syndrome similar to the phenotype of mice harboring a complete deletion of p53 (Villin-Cre; p53 FL/FL) (**Figure 5F in the revised manuscript; please see below**).

[REDACTED]

Accordingly, we observed that the expression of Clu mRNA was significantly suppressed in intestinal epithelial cells of VillinCre; LSL-p53 25,26/FL mice after 12 Gy SBI (**Rebuttal Figure 14**). These preliminary data are only included in the rebuttal letter to share with the Reviewer because we are planning to publish them in a follow-up paper. Together, our findings demonstrate the transcriptional activity of p53 in response to acute DNA damage plays a critical role in promoting the emergence Clu+ revSCs in the irradiated intestinal epithelium and protecting mice from radiation-induced acute GI syndrome.

Rebuttal Figure 13. [REDACTED]

Rebuttal Figure 14. p53 transcriptional activity is required to induce Clu during intestinal regeneration. a) Representative images from sm-FISH of *Clu* expression in mice with p53 WT (*Villin^{Cre}; p53^{FL/+}*), p53 KO (*Villin^{Cre}; p53^{FL/FL}*) or p53 TAD mutants (*Villin^{Cre}; p53^{LSL-25,26/FL}*) intestinal epithelium in non-irradiated (NR) 60h after irradiation (IR). Scale bar=50µm. **b)** Quantification of (A).

2. The authors only focused on the p53 involvement in the regeneration after lethal IR dosages. What happens when the IR dosages are lower? This will help to understand the mechanism of such protective role.

We thank the Reviewer for this question about the impact of a lower radiation dose. We performed lineage tracing experiments to assess the contribution of Clu+ epithelial cells to intestinal regeneration following both 10 and 14 Gy SBI. It has been shown that 10 Gy and 14 Gy reduce the number of proliferating crypts by ~50% and 95%, respectively. Our data from lineage tracing using Clu-CreER; LSL-tdTomato mice showed that there were significantly more intestinal crypts regenerated through Clu+ revSCs following 14 Gy compared to 10 Gy (**Figure S2 in the revised manuscript**). Together, these results indicate that Clu-mediated regeneration is more predominant following radiation doses that cause severe damage to the intestinal epithelium.

3. In the context of physiological and clinical relevance, p53 is commonly mutated in intestinal cancers. the authors should extend their analysis to p53 mutant mouse models such as R175H, R248W knock-in mice.

We agree with the Reviewer that an interesting and important question raised from our study is to understand the radiation response of premalignant and malignant intestinal epithelial cells that harbor various dominant-negative p53 mutations (such as R175H and R248W). However, we haven't experimentally addressed this comment for several reasons that we explain below:

1. Scope of the study: Despite the interesting comment raised by the reviewer, the focus of the current manuscript is on dissecting the mechanisms by which p53 promotes the regeneration of normal intestinal epithelial cells following severe radiation injury. We appreciate the reviewer for motivating us to expand our study in other models, but we believe that this comment is out of the scope of the current manuscript.

2. Conceptual interpretation: Most of these p53 hot spot mutations are missense mutations within the DNA-binding core domain that affect the p53 transcriptional activity (Kern et al., Science 1992; PMID: 1589764). Whether these mutations affect p53 contact with DNA (contact mutations - R248W) or p53 conformation (structural mutations-R175H), all of them abrogate the wild-type

tumor suppressor function of p53. We appreciate the reviewer bringing up this point because, indeed, our study emphasizes the importance of these transactivation domains as p53 TAD mutants are also more sensitive to GI syndrome similar to p53 full knockouts. Importantly, it has also been shown that despite the loss of p53 canonical transcriptional function, these point mutations endow p53 with a new oncogenic gain of function activities to promote cancer (Olive et al., Cell 2004; PMID: 15607980) (Lang et al., Cell 2004; PMID: 15607981) (Song et al., Nature Cell Biology 2007; PMID: 17417627). Thus, it would be very difficult to translate and interpret our findings of p53-mediated radioprotection in normal tissue into these cancer models.

3. Technical limitation: To address this comment we would have to acquire and establish these mouse models in our laboratory. This would require extending the revision process beyond a reasonable time frame.

Finally, we would also like to emphasize that we still believe that our findings have important physiological and clinical relevance. Radiotherapy is one of the main standards of care for cancer patients. However, the efficacy of these treatments is very limited by the toxicity generated in the normal surrounding tissues. Although our study focuses on high-dose radiation, the expanding use of hypofractionation makes these radiation doses more clinically relevant and we believe that our discoveries provide novel mechanistic insight into tissue regeneration and will inspire new strategies to overcome radiation-induced toxicities in cancer patients.

REVIEWERS' COMMENTS

Reviewer #2 (Remarks to the Author):

Overall, the authors thoroughly addressed comments and concerns from the reviewers. However, the authors have not improved the images of organoids that represented lineage tracing of Clu+ cells in Fig.3F and 4G. The distinction between the yellow signals (Clu+ tracing cells) emitted on the luminal side and the epithelial cells in crypts remains unclear. Notably, it would be challenging for readers unfamiliar with organoids to understand the data in the current version of those figures. I observed yellow signals of entire organoids in Fig. 4G (0 Gy, Nutlin-3). Are these signals expressed in epithelium? Or are they released into the lumen? If the former, what causes the rise in Clu+ cells following Nutlin-3 treatment? Even if the mechanism underlying this phenomenon is unclear, it should be stated in the text.

The study represents an essential advance in the function of p53 on the revival stem cells during intestinal epithelial cell reprogramming upon GI injury, but given that mysteries still remain regarding the molecular mechanism of p53. In the current version of the manuscript, the conceptual advance of this study is limited because the authors have already reported that VillinCre-p53 FL/- mouse remained sensitive to irradiation (Kirsch et al. Science 2010). YAP signaling has also been a well-described signaling pathway in the reprogramming process in the intestine. Therefore, it would be beneficial to add a brief discussion regarding the connection between p53 and YAP signaling on this open subject as a focus for future work, as well as the analysis of YAP-related genes on single-cell RNA-seq (Rebuttal Fig.5,6) in the manuscript.

After the authors address the weakness mentioned above with further small in vitro experiments and discussion, I believe the article will become appropriate for publication in Nature Communications and be seen by a wider audience.

Reviewer #3 (Remarks to the Author):

The authors addressed my concerns satisfactorily, but with the data that will not be included in this manuscript. It is up to the editor to decide whether some of the data in the rebuttal letter should be included in this manuscript.

Concern#1 from Reviewer 2

Overall, the authors thoroughly addressed comments and concerns from the reviewers. However, the authors have not improved the images of organoids that represented lineage tracing of Clu+ cells in Fig.3F and 4G. The distinction between the yellow signals (Clu+ tracing cells) emitted on the luminal side and the epithelial cells in crypts remains unclear. Notably, it would be challenging for readers unfamiliar with organoids to understand the data in the current version of those figures. I observed yellow signals of entire organoids in Fig. 4G (0 Gy, Nutlin-3). Are these signals expressed in epithelium? Or are they released into the lumen? If the former, what causes the rise in Clu+ cells following Nutlin-3 treatment? Even if the mechanism underlying this phenomenon is unclear, it should be stated in the text.

We have included an illustration and images of individual fluorescent channels of the organoid lineage tracing data in **Supplementary Figures 3 a and b** of the revised manuscript to clarify the design and interpretation of the organoid experiments.

Supplementary Figure 3

Supplementary Figure 3. Clu⁺ cells contribute to the regeneration of the small intestines following various doses of sub-total body irradiation.

(A) Schematic representing the lineage tracing model from Clu⁺ cells after IR damage in intestinal organoids *in vitro*. (Created with [BioRender.com](https://www.biorender.com)).

(B) Individual immunofluorescence channels (DAPI-blue, tdTOMATO-yellow, and Lgr-GFP-green) from Figure 3f.

We also revised the text in the Results as the following:

“To determine the impact of acute p53 inhibition after irradiation on the regeneration of intestinal crypts via Clu⁺ revSCs, we established a radiation injury model in intestinal organoids from *Clu^{Cre-ERT2/+}; Rosa26^{LSL-tdTomato/+}; Lgr5^{DTR-GFP}* mice to enable *in vitro* lineage tracing experiments (**Supplementary Fig. 3a**). These organoids were treated with 4-hydroxy tamoxifen (4-OHT) to label the Clu⁺ revSCs and their offspring 2 hours before irradiation with 0, 3 or 5 Gy. Sixteen hours after irradiation, organoids were passaged and treated with either vehicle alone (DMSO) or pifithrin-alpha (PF-a), a small molecule inhibitor of p53²². Five days after passage, we examined the regeneration of Lgr5⁺ cells labeled by GFP (green color) from the offspring of Clu⁺ revSCs labeled by tdTomato (yellow color) (**Fig. 3e, Supplementary Fig. 3b**).”

Concern#2 from Reviewer 2

The study represents an essential advance in the function of p53 on the revival stem cells during intestinal epithelial cell reprogramming upon GI injury, but given that mysteries still remain regarding the molecular mechanism of p53. In the current version of the manuscript, the conceptual advance of this study is limited because the authors have already reported that VillinCre-p53 FL/- mouse remained sensitive to irradiation (Kirsch et al. Science 2010). YAP signaling has also been a well-described signaling pathway in the reprogramming process in the intestine. Therefore, it would be beneficial to add a brief discussion regarding the connection between p53 and YAP signaling on this open subject as a focus for future work, as well as the analysis of YAP-related genes on single-cell RNA-seq (Rebuttal Fig.5,6) in the manuscript.

We have included Rebuttal Figures 5 and 6 as **Supplementary Figure 2** of the revised manuscript.

Supplementary Figure 2

Supplementary Figure 2. Yap signaling is activated in revival stem cells (revSC) after intestinal irradiation.

(A) UMAP showing the transcriptional expression of downstream Yap target genes (Supplementary Data 2) across all intestinal epithelial cells (left) and in NR, day 2 and day 3 after irradiation (right).

(B) UMAP representation of single-cell transcriptome profile of intestinal epithelial cells from *Lats1^{fl/fl} Lats2^{fl/fl}* (WT) and *Lrig-CreERT2- Lats1^{fl/fl} Lats2^{fl/fl}* (*Lats 1/2cKO*) 7 days after tamoxifen induction. (Data obtained from *Cheung et al., Cell Stem Cell 2020*).

(C) Expression of revSCs associated genes (*Clu*, *Ly6a*, *Anxa1*, *Cdkn1a*) and p53 target genes (*Trp53*, *Phlda3*, *Bax*, *Atg9b*) in WT or *Lats 1/2cKO* intestinal epithelial cells.

We also revised the text in the Results as the following:

“Indeed, we examined the transcriptional targets of Yap and found that these genes are also enriched in the revSC population after 12 Gy SBI (Supplementary Fig. 2a).”

“To further investigate the potential relationship between Yap and p53 signaling, we revisited a scRNA-seq data set published by *Cheung et al., 2020*²¹. In this study, the authors deleted the *Lats1/2* kinases in the mouse intestinal stem cells to induce Yap hyperactivation. Using this data set, we examined the expression of revSCs gene markers in addition to well-known transcriptional

targets of p53. We found that hyperactivation of Yap via Lats1/2 knockout upregulated both sets of genes in intestinal stem cells, supporting the notion that Yap signaling cooperates with p53 pathway activation to induce revSCs (**Supplementary Fig. 2b,c**)."